# Improving Classifier-Free Guidance of Flow Matching via Manifold Projection

**Jian-Feng Cai** [1 2]   **Haixia Liu** [3]   **Zhengyi Su** [4]   **Chao Wang** [4 †]

## Abstract

Classifier-free guidance (CFG) is a widely used technique for controllable generation in diffusion and flow-based models. Despite its empirical success, CFG relies on a heuristic linear extrapolation that is often sensitive to the guidance scale. In this work, we provide a principled interpretation of CFG through the lens of optimization. We demonstrate that the velocity field in flow matching corresponds to the gradient of a sequence of smoothed distance functions, which guides latent variables toward the scaled target image set. This perspective reveals that the standard CFG formulation is an approximation of this gradient, where the prediction gap, the discrepancy between conditional and unconditional outputs, governs guidance sensitivity. Leveraging this insight, we reformulate the CFG sampling as a homotopy optimization with a manifold constraint. This formulation necessitates a manifold projection step, which we implement via an incremental gradient descent scheme during sampling. To improve computational efficiency and stability, we further enhance this iterative process with Anderson Acceleration without requiring additional model evaluations. Our proposed methods are training-free and consistently refine generation fidelity and robustness to the guidance scale. We validate their effectiveness across diverse benchmarks, demonstrating significant improvements on large-scale models such as DiT-XL-2-256, Flux, and Stable

Authors are listed in alphabetical order. [1]Department of Mathematics, The Hong Kong University of Science and Technology, Hong Kong, China; [2]IAS Center for AI for Scientific Discoveries, The Hong Kong University of Science and Technology, Hong Kong, China; [3]School of Mathematics and Statistics & Institute of Interdisciplinary Research for Mathematics and Applied Science & Hubei Key Laboratory of Engineering Modeling and Scientific Computing, Huazhong University of Science and Technology, Wuhan, China; [4]Department of Statistics and Data Science, Southern University of Science and Technology, Shenzhen, China. Correspondence to: Chao Wang <wangc6@sustech.edu.cn>.

*Proceedings of the $43^{rd}$ International Conference on Machine Learning*, Seoul, South Korea. PMLR 306, 2026. Copyright 2026 by the author(s).

Diffusion 3.5. Code is available at `https://github.com/LeonSuZhengYi/CFG-MP`.

## 1. Introduction

Diffusion and flow methods now lead the field for producing top-tier images and videos (Ho et al., 2020; Song et al., 2021; Lipman et al., 2023; Liu et al., 2023). Notably, Flow Matching (FM), which trains a neural network to approximate a specified vector field that transports random noise to data samples, has become a simulation-free training strategy for flow-based generative models, delivering excellent numerical stability, lower computational overhead, and strong modeling flexibility.

In conditional generation tasks, such as class-conditional or text-to-image synthesis, Classifier-Free Guidance (CFG) (Ho & Salimans, 2021) has become the dominant mechanism for steering generation toward desired conditions. CFG operates by linearly extrapolating between unconditional and conditional model predictions, controlled by a scalar guidance scale $w$. Despite its simplicity and empirical success, CFG exhibits a well-known trade-off: small values of $w$ result in weak conditional alignment, whereas large values often cause oversaturation, artifacts, or loss of diversity. In practice, the choice of $w$ is highly sensitive and model-dependent, and is typically selected through ad-hoc tuning. This sensitivity highlights a limitation of CFG in complex sampling scenarios.

To address this limitation, various refinements have been proposed. In the diffusion context, methods proposed in Lugmayr et al. (2022); Bai et al. (2025); Chung et al. (2025); Wang et al. (2025); Sadat et al. (2025); Karras et al. (2024); Zheng & Lan (2024) focus on stabilizing the guidance process. Similarly, flow-specific variants (Fan et al., 2025; Saini et al., 2025) have sought to improve sampling quality. Detailed descriptions of these related studies are provided in Related works (Section A). Notably, Wang et al. (2025) suggest that reducing the prediction gap, the discrepancy between unconditional and conditional predictions, can significantly enhance sampling performance. While empirically effective, existing approaches leave an important gap. On one hand, many methods are diffusion-centric, relying on discretization or noise-schedule properties that do not

directly transfer to flow matching, where sampling follows a continuous-time ODE defined by a velocity field. On the other hand, the connection between prediction-gap reduction and improved sampling remains largely unexplained from a theoretical standpoint. More precisely, CFG lacks a theoretical interpretation with three key remaining questions. First, what is the objective function that the standard CFG extrapolation actually approximates when applied to flow-matching models? Second, does there exist a mathematically optimal guidance scale $w$, and if so, how can it be defined and estimated? Third, why do recent empirical methods that reduce the discrepancy between conditional and unconditional predictions, often referred to as the "prediction gap", consistently improve generation quality and robustness? Addressing these questions requires moving beyond heuristic interpretations of CFG toward an optimization-based understanding.

In this work, we bridge this gap by reinterpreting conditional flow-matching sampling through the lens of optimization. We show that the ideal conditional velocity field can be expressed as the gradient of a sequence of regularized distance objectives, where each objective attracts noisy latent variables toward a scaled conditional image set. From this perspective, standard CFG emerges as an approximation to this gradient field. Crucially, this interpretation allows us to formally define an optimal guidance scale $w^*$ as the minimizer of the approximation error. Building on this result, we decompose the approximation error as the sum of an unavoidable model error term and a scaled prediction gap. This decomposition provides a theoretical explanation that eliminating the prediction gap improves robustness and reduces sensitivity to the guidance scale.

Building on this analysis, we introduce a manifold constraint to eliminate the prediction gap and propose a novel CFG-based sampling method via a homotopy optimization process under this constraint. This leads to a simple yet effective sampling scheme, termed CFG-MP, which incorporates an iterative projection toward this manifold in each step of CFG sampling. To address the potential computational overhead of iterative projection, we utilize Anderson Acceleration, yielding CFG-MP+, which significantly improves convergence speed and stability in practice.

Our main contributions are summarized as follows:

- **Optimization-based interpretation of CFG sampling process.** We show that the conditional velocity field in flow matching corresponds to the gradient of a regularized distance objective. This interpretation allows us to formally define the optimal guidance scale as the optimal approximation of the true conditional gradient.

- **Theoretical analysis of guidance sensitivity.** We de-

rive an error decomposition for CFG-based sampling that explicitly characterizes the interplay between guidance scale sensitivity and the prediction gap between conditional and unconditional model outputs.

- **CFG-MP: a manifold projection sampling method.** We formulate conditional sampling as a constrained optimization problem and propose CFG-MP, an incremental gradient descent scheme that reduces the prediction gap during sampling without modifying or retraining the underlying generative model.

- **CFG-MP+: an acceleration of CFG-MP.** We integrate Anderson Acceleration into CFG-MP to stabilize and accelerate convergence, substantially enlarging the practical stability region of iterative projection methods. Extensive experiments on class-to-image and text-to-image tasks demonstrate that our methods outperform state-of-the-art CFG variants across multiple models and datasets.

**Conflict of Interest Disclosure.** The authors declare no financial conflicts of interest related to this work.

## 2. Preliminary

**Conditional and unconditional flow matching.** We begin with a review of conditional flow matching (CFM). Let $\mathcal{Y}$ denote the space of conditioning signals (e.g., text prompts), CFM aims to learn a velocity field $v_\theta : [0, 1] \times \mathbb{R}^d \times \mathcal{Y} \to \mathbb{R}^d$ that is represented by a neural network with parameters $\theta$. Given $y \in \mathcal{Y}$, the velocity field $v_\theta(\cdot, \cdot, y)$ generates a probability density path connecting the Gaussian distribution $p_0 := \mathcal{N}(0, I)$ and the conditional data distribution $p_1^y := p_1(\cdot|y)$ by an ordinary differential equation (ODE):

$$\tfrac{d}{dt}x_t = v_\theta(t, x, y), \ x_0 \sim p_0, \ x_1^y \sim p_1^y, \ t \in (0, 1). \quad (2.1)$$

In the training procedure, we sample $x_t = (1 - t)x_0 + tx_1^y$ with $t \in (0, 1)$. The neural network $v_\theta(\cdot, \cdot, y)$ is then trained by minimizing the following loss (Lipman et al., 2023):

$$\mathcal{L}^y(\theta) = \mathbb{E}_{t, x_0, x_1^y} \left[ \|v_\theta(t, x_t, y) - (x_1^y - x_0)\|^2 \right]. \quad (2.2)$$

Once trained, new samples are generated by solving the ODE (2.1) with numerical solvers (Lipman et al., 2024).

Similarly, for unconditional flow matching (UFM), we aim to learn a velocity field $v_\theta(\cdot, \cdot, \emptyset) : [0, 1] \times \mathbb{R}^d \to \mathbb{R}^d$ with the unconditional data distribution $p_1^\emptyset := p_1(\cdot|\emptyset)$ by an ODE:

$$\tfrac{d}{dt}x_t = v_\theta(t, x, \emptyset), \ x_0 \sim p_0, \ x_1 \sim p_1^\emptyset, \ t \in (0, 1).$$

And the training loss is similarly defined as:

$$\mathcal{L}^\emptyset(\theta) = \mathbb{E}_{t, x_0, x_1^\emptyset} \left[ \|v_\theta(t, x_t, \emptyset) - (x_1^\emptyset - x_0)\|^2 \right]. \quad (2.3)$$

**Classifier-free guidance for flow matching.** To manage the CFM and UFM simultaneously, in the CFG framework (BlackForestLabs et al., 2025; Ho & Salimans, 2021), we employ a single neural network $v_\theta$ to represent both $v_\theta(\cdot, \cdot, y)$ and $v_\theta(\cdot, \cdot, \emptyset)$ using shared parameters $\theta$. Consequently, we minimize the following joint objective function:

$$\mathcal{L}^{\text{cfg}}(\theta) = \mathbb{E}_{c,t,x_0,x_1^c} \left[ \|v_\theta(t, x_t, c) - (x_1^c - x_0)\|^2 \right]$$
$$= q\mathcal{L}^y(\theta) + (1-q)\mathcal{L}^\emptyset(\theta), \quad (2.4)$$

where $c = y$ with probability $q$ and $c = \emptyset$ with probability $1 - q$, and $q \in (0, 1)$ is a training hyperparameter. Since these two fields are optimized simultaneously, the learned velocity field $v_\theta(t, x, y)$ may not yield the optimal trajectory for the sampling process, as shown in the experiments of Ho & Salimans (2021). To mitigate this, CFG utilizes an extrapolation of these fields to approximate the enhanced conditional velocity field during sampling:

$$v_\theta^{\text{cfg}}(t, x, y) := v_\theta(t, x, \emptyset) + w(v_\theta(t, x, y) - v_\theta(t, x, \emptyset)), \quad (2.5)$$

where $w \geq 1$ is a hyperparameter for the guidance scale, chosen empirically. In the sampling process of CFG, we solve the ODE $\frac{d}{dt}x_t = v_\theta^{\text{cfg}}(t, x_t, y)$. In Section 3, we will provide a comprehensive theoretical analysis of this method and introduce further improvements to CFG sampling techniques.

## 3. Methodology

### 3.1. CFG under an optimization perspective

Here, we recast the training and sampling processes of CFG from an optimization-centric perspective. During training, we derive the ideal velocity fields by analyzing the CFG objective in the function space despite the neural network parameterization. For sampling, we reformulate the flow ODE as a homotopy optimization process, aiming to minimize the distance function relative to the target image set.

**The minimizer of the CFG training functional.** In the CFG training process, the objective ideally entails seeking the target velocity fields within an infinite-dimensional function space, rather than being constrained to the finite-dimensional parameterization in (2.4). The counterparts of (2.2) and (2.3) over the continuous velocity field space $C((0, 1) \times \mathbb{R}^d, \mathbb{R}^d)$ are:

$$\mathcal{L}^y(f) := \mathbb{E}_{t,x_0 \sim p_0, x_1^y \sim p_1^y} \left[ \|f(t, x_t) - (x_1^y - x_0)\|^2 \right],$$
$$\mathcal{L}^\emptyset(g) := \mathbb{E}_{t,x_0 \sim p_0, x_1^\emptyset \sim p_1^\emptyset} \left[ \|g(t, x_t) - (x_1^\emptyset - x_0)\|^2 \right].$$

Then $\mathcal{L}^{\text{cfg}}(f, g) := q\mathcal{L}^y(f) + (1-q)\mathcal{L}^\emptyset(g)$ is a functional in the product space $\left( C((0, 1) \times \mathbb{R}^d, \mathbb{R}^d) \right)^2$ corresponding to the CFG training loss.

Now, we present the explicit expressions of the ideal velocity fields, i.e., the minimizers of these functionals. Define

$$v_{t,y}^*(x) := \frac{1}{1-t}(\mathbb{E}_{\tilde{x}_1 \sim p_1^y(\cdot|x,t)} [\tilde{x}_1] - x), \quad (3.1)$$
$$v_{t,\emptyset}^*(x) := \frac{1}{1-t}(\mathbb{E}_{\tilde{x}_1 \sim p_1^\emptyset(\cdot|x,t)} [\tilde{x}_1] - x), \quad (3.2)$$

where $p_1^y(\cdot|x, t)$ and $p_1^\emptyset(\cdot|x, t)$, given in (B.1) and (B.3) respectively, are two modulated posterior data distributions. Then, we have:

**Theorem 3.1.** *Assume* $v_{\cdot,y}^*(\cdot), v_{\cdot,\emptyset}^*(\cdot) \in C((0, 1) \times \mathbb{R}^d, \mathbb{R}^d)$ *and* $\mathcal{L}^y(v_{\cdot,y}^*), \mathcal{L}^\emptyset(v_{\cdot,\emptyset}^*) < \infty$. *Then, the global minimizers for the functionals* $\mathcal{L}^y$ *and* $\mathcal{L}^\emptyset$ *over the space* $C((0, 1) \times \mathbb{R}^d, \mathbb{R}^d)$ *are* $v_{\cdot,y}^*(\cdot)$ $v_{\cdot,\emptyset}^*(\cdot)$ *respectively. Moreover, the pair* $(v_{\cdot,y}^*(\cdot), v_{\cdot,\emptyset}^*(\cdot))$ *is the global minimizer of the functional* $\mathcal{L}^{\text{cfg}}(\cdot, \cdot)$ *over* $\left( C((0, 1) \times \mathbb{R}^d, \mathbb{R}^d) \right)^2$.

The proof of Theorem 3.1 is deferred to Appendix B.

**The CFG sampling as a homotopy optimization.** In the following, we show that the ideal velocity field is the gradient of a regularized distance function. Let $\mathcal{K}_y$ and $\mathcal{K}_\emptyset$ be the sets of images conditioned on $y$ and the unconditional images, respectively. Define the following distance functions

$$\text{dist}_{\mathcal{K}_y}(x) := \min_{x_1 \in \mathcal{K}_y} \|x - x_1\|,$$
$$\text{dist}_{\mathcal{K}_\emptyset}(x) := \min_{x_1 \in \mathcal{K}_\emptyset} \|x - x_1\|,$$

and we employ the Log-Sum-Exp technique (Permenter & Yuan, 2024) to construct smooth approximations of these squared distance functions:

**Definition 3.2.** For any $x \in \mathbb{R}^d$, the smoothed squared-distance function to $\mathcal{K}_y$ and $\mathcal{K}_\emptyset$ with a smoothing parameter $\sigma > 0$ are defined as:

$$\text{dist}_{\mathcal{K}_y}^2(x, \sigma) := -2\sigma^2 \log \left( \mathbb{E}_{x_1 \sim p_1^y} \left[ \exp \left( -\frac{\|x_1 - x\|^2}{2\sigma^2} \right) \right] \right),$$
$$\text{dist}_{\mathcal{K}_\emptyset}^2(x, \sigma) := -2\sigma^2 \log \left( \mathbb{E}_{x_1 \sim p_1^\emptyset} \left[ \exp \left( -\frac{\|x_1 - x\|^2}{2\sigma^2} \right) \right] \right),$$

where $p_1^y$ and $p_1^\emptyset$ are the density functions supported on $\mathcal{K}_y$ and $\mathcal{K}_\emptyset$ respectively.

It can be verified that $\min_{x_1 \in \mathcal{K}_y} \|x - x_1\|^2 \leq \text{dist}_{\mathcal{K}_y}^2(x, \sigma) \leq \max_{x_1 \in \mathcal{K}_y} \|x - x_1\|^2$, and $\text{dist}_{\mathcal{K}_y}^2(x, \sigma) \to \text{dist}_{\mathcal{K}_y}^2(x)$ as $\sigma \to 0$, so the smoothed squared-distance is indeed a smooth approximation of the squared distance function. Moreover, when $p_1^y$ and $p_1^\emptyset$ are arbitrary distributions, $\text{dist}_{\mathcal{K}_y}^2(x, \sigma)$ and $\text{dist}_{\mathcal{K}_\emptyset}^2(x, \sigma)$ are still well-defined; in this case, they can be viewed as the smooth approximations of negative log density $-\log(p_1^y)$ and $-\log(p_1^\emptyset)$ up to an affine transform.

The ideal velocity fields $v_{t,y}^*(x)$ and $v_{t,\emptyset}^*(x)$ in (3.1) and (3.2) are closely related to the gradients of these smoothed distance functions, as established in the following theorem:

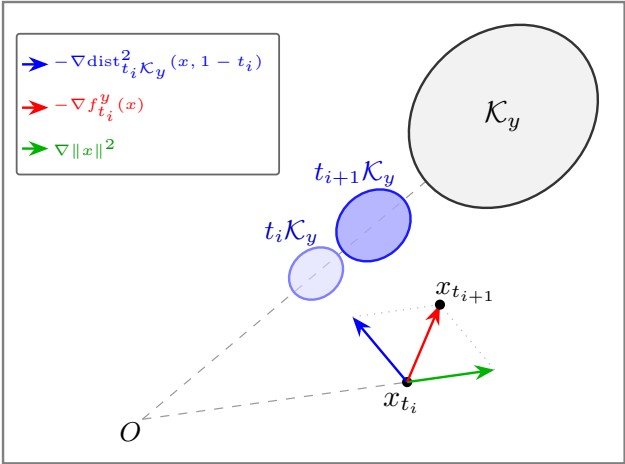

*Figure 1.* Illustration of the homotopy optimization process (3.3).

**Theorem 3.3.** *We have the following relations:*

$$v_{t,y}^*(x) = -\frac{1}{2t(1-t)}\nabla_x \left\{ \text{dist}_{t\mathcal{K}_y}^2(x, 1-t) - (1-t)\|x\|^2 \right\},$$
$$v_{t,\emptyset}^*(x) = -\frac{1}{2t(1-t)}\nabla_x \left\{ \text{dist}_{t\mathcal{K}_\emptyset}^2(x, 1-t) - (1-t)\|x\|^2 \right\}.$$

The proof of Theorem 3.3 can be found in Appendix B. With this theorem, we can now interpret CFG sampling as a homotopy optimization process (Lin et al., 2023; Iwakiri et al., 2022). Specifically, the sampling process should be governed by the ODE $\frac{d}{dt}x_t = v_{t,y}^*(x_t)$. Its Euler discretization with $\Delta t = t_{i+1} - t_i$ is

$$x_{i+1} = x_i + \Delta t \cdot v_{t_i,y}^*(x_i)$$
$$= x_i - \frac{\Delta t}{2t_i(1-t_i)} \cdot \nabla f_{t_i}^y(x_i), \qquad (3.3)$$

where the second equality follows from Theorem 3.3 and the following definition:

$$f_t^y(x) := \text{dist}_{t\mathcal{K}_y}^2(x, 1-t) - (1-t)\|x\|^2. \qquad (3.4)$$

Since $f_t^y \to \text{dist}_{\mathcal{K}_y}^2$ as $t \to 1$, the sampling process can be regarded as a homotopy optimization to minimize $\text{dist}_{\mathcal{K}_y}(x)$, where each step $t_i$ performs a single-step gradient descent on $f_t^y$. Moreover, from (3.3) and (3.4), we get the following geometric intuition: in the sampling, $x_t$ is not only attracted by $t\mathcal{K}_y$ but also pushed away from the origin; see Figure 1.

### 3.2. The approximation error of the ideal velocity field

In CFG, we use $v_\theta^{\text{cfg}}(t, x, y)$ in (2.5) to approximate the ideal velocity field $v_{t,y}^*(x)$. This approximation error is a quadratic function of $w$:

$$\|v_{t,y}^*(x) - v_\theta(t, x, \emptyset) - w(v_\theta(t, x, y) - v_\theta(t, x, \emptyset))\|^2.$$

Minimizing this quadratic function yields the optimal guidance scale

$$w^* = \frac{\langle v_\theta(t, x, y) - v_\theta(t, x, \emptyset), v_{t,y}^*(x) - v_\theta(t, x, \emptyset) \rangle}{\|v_\theta(t, x, y) - v_\theta(t, x, \emptyset)\|^2}.$$

While $w^*$ is computationally intractable in practice due to the unknown ideal velocity $v_{t,y}^*(x)$, we can nonetheless decompose the approximation error as:

**Theorem 3.4.** *The approximation error* $\|v_\theta^{\text{cfg}}(t, x, y) - v_{t,y}^*(x)\|^2$ *can be decomposed as*

$$\|v_\theta^{\text{cfg}}(t, x, y) - v_{t,y}^*(x)\|^2 = \|v_\theta^{\text{cfg}*}(t, x, y) - v_{t,y}^*(x)\|^2$$
$$+ (w^* - w)^2 \|v_\theta(t, x, y) - v_\theta(t, x, \emptyset)\|^2,$$

*where* $v_\theta^{\text{cfg}*}$ *is the optimal CFG extrapolation with* $w^*$, *i.e.,*

$$v_\theta^{\text{cfg}*}(t, x, y) := v_\theta(t, x, \emptyset) + w^*(v_\theta(t, x, y) - v_\theta(t, x, \emptyset)).$$

The proof of Theorem 3.4 can be found in Appendix B. In the error decomposition in Theorem 3.4, the first term represents the intrinsic model error associated with the CFG extrapolation (2.5), which remains intractable and irreducible. The second term is proportional to $(w^* - w)^2$, governing the sensitivity to guidance scale error. Notably, the sensitivity coefficient is defined as $\|v_\theta(t, x, y) - v_\theta(t, x, \emptyset)\|^2$. Following the terminology used in recent diffusion research (Wang et al., 2025), we refer to this as the prediction gap. Crucially, a larger prediction gap exacerbates the sensitivity of the approximation error to the guidance scale. Consequently, we focus on eliminating the prediction gap rather than searching for an optimal $w$ to mitigate the approximation error. To our knowledge, this constitutes the first theoretical analysis of the CFG approximation error through the lens of the prediction gap.

### 3.3. CFG-MP: CFG sampling with manifold projection

To mitigate the impact of the prediction gap, we impose a manifold constraint $\mathcal{M}_t \subset \mathbb{R}^d$, defined as:

$$\mathcal{M}_t := \{z \mid v_\theta(t, z, y) = v_\theta(t, z, \emptyset)\}.$$

We then reformulate the gradient descent iteration from the homotopy optimization in (3.3) by incorporating a projection onto $\mathcal{M}_{t_{i+1}}$ prior to the next time step $t_{i+1}$. The resulting update rule is:

$$x_{i+1} = \text{Proj}_{\mathcal{M}_{t_{i+1}}}\left(x_i - \frac{\Delta t}{2t_i(1-t_i)}\nabla f_{t_i}^y(x_i)\right) \qquad (3.5)$$
$$\approx \text{Proj}_{\mathcal{M}_{t_{i+1}}}(x_i + \Delta t \cdot v_\theta^{\text{cfg}}(t, x_i, y)).$$

This formulation effectively implements Projected Gradient Descent (PGD), where $\text{Proj}_{\mathcal{S}}(x) := \arg\min_z \{\|z - x\| \text{ s.t. } z \in \mathcal{S}\}$ denotes the Euclidean projection operator.

**The implementation of $\text{Proj}_{\mathcal{M}_{t_{i+1}}}$.** The absence of a closed-form solution for $\text{Proj}_{\mathcal{M}_{t_{i+1}}}$ necessitates an iterative approximation. We first introduce a proxy manifold defined by the ideal velocity fields,

$$\mathcal{M}'_t := \{z \mid v^*_{t,y}(z) = v^*_{t,\emptyset}(z)\}.$$

This distinction is important: an arbitrary learned vector field $v_\theta$ does not necessarily admit a potential function whose stationary points exactly characterize $\mathcal{M}_t$. In contrast, Theorem 3.3 shows that the ideal fields $v^*_{t,y}$ and $v^*_{t,\emptyset}$ are gradients of smoothed distance functions. Hence $\mathcal{M}'_t$ can be characterized as the stationary set of a well-defined potential function $F_t(x)$:

$$\mathcal{M}'_t = \{x \mid 0 = \nabla F_t(x) := \nabla \frac{1}{2t(1-t)}(f^y_t(x) - f^{\emptyset}_t(x))\},$$

where $f^y_t(x)$ is defined in (3.4) and

$$f^{\emptyset}_t(x) := \text{dist}^2_{t\mathcal{K}_{\emptyset}}(x, 1-t) - (1-t)\|x\|^2. \quad (3.6)$$

Thus, in the derivation, we use $\mathcal{M}_t \approx \mathcal{M}'_t$ and implement the projection through this proxy manifold. Considering the geometric interpretation:

$$\begin{aligned}
2t(1-t)F_t(x) &= f^y_t(x) - f^{\emptyset}_t(x) \\
&= \text{dist}^2_{t\mathcal{K}_y}(x, 1-t) - \text{dist}^2_{t\mathcal{K}_{\emptyset}}(x, 1-t) \\
&\approx \text{dist}^2_{t\mathcal{K}_y}(x) - \text{dist}^2_{t\mathcal{K}_{\emptyset}}(x),
\end{aligned}$$

Since our primary goal is to satisfy the condition $y$ (i.e., reduce $\text{dist}^2_{t\mathcal{K}_y}(x)$), minimizing the composite objective $F_t(x)$ is the appropriate strategy. Maximizing $F_t(x)$ would inevitably increase the distance to $\mathcal{K}_y$, which is counterproductive to the sampling process.

To minimize the composite objective $F_t(x)$, we employ an incremental gradient descent approach (Gürbüzbalaban et al., 2019; Bertsekas, 2011). Instead of updating based on the full gradient of $F_t(x)$ simultaneously, we partition the objective into its unconditional and conditional components and perform sequential updates to the state. Specifically, we define the two-step incremental update as:

$$\begin{cases} z_k = x_k + a\nabla_x \frac{f^{\emptyset}_t(x_k)}{2t(1-t)}, \\ x_{k+1} = z_k - a\nabla_x \frac{f^y_t(z_k)}{2t(1-t)}. \end{cases} \quad (3.7)$$

By Theorem 3.3, these gradients can be expressed in terms of the ideal velocity fields, giving $z_k = x_k - av^*_{t,\emptyset}(x_k)$ and $x_{k+1} = z_k + av^*_{t,y}(z_k)$. Since $v^*$ is computationally intractable during sampling, we use the trained network as an implementable approximation, namely $v_\theta \approx v^*$. This substitutes the ideal updates with $z_k = x_k - av_\theta(t, x_k, \emptyset)$ and $x_{k+1} = z_k + av_\theta(t, z_k, y)$. By setting $a = \frac{1}{2}\Delta t$ (see Appendix C.2.1), we arrive at the integrated iterative scheme $x_{k+1} = G(x_k, t)$, defined as:

$$\begin{aligned}
G(x, t) :=& x - \tfrac{1}{2}\Delta t \cdot v_\theta(t, x, \emptyset) \\
&+ \tfrac{1}{2}\Delta t \cdot v_\theta(t, x - \tfrac{1}{2}\Delta t \cdot v_\theta(t, x, \emptyset), y).
\end{aligned}$$

*Remark* 3.5. It is worth noting that if $\mathcal{M}_t = \emptyset$, $G(x, t)$ continues to function as an optimization step that minimizes the potential $F_t(x)$. Therefore, although the prediction gap may not be strictly eliminated, it is significantly reduced, thereby ensuring that $x_{t_{i+1}}$ maintains a minimal deviation from the ideal trajectory at the next time step.

*Remark* 3.6. In the context of incremental methods, the order of updates matters. We could alternatively perform the conditional step first, followed by the unconditional step. This yields the scheme $x_{k+1} = H(x_k, t)$:

$$\begin{aligned}
H(x, t) :=& x + \tfrac{1}{2}\Delta t \cdot v_\theta(t, x, y) \\
&- \tfrac{1}{2}\Delta t \cdot v_\theta(t, x + \tfrac{1}{2}\Delta t \cdot v_\theta(t, x, y), \emptyset).
\end{aligned}$$

Empirically, the operator $G(x, t)$ exhibits more stable convergence behavior and superior performance over $H(x, t)$, even in regimes where both methods theoretically converge. Consequently, we utilize $G(x, t)$ for all subsequent derivations and experiments.

The complete method is summarized in Algorithm 1 and termed CFG-MP.

---

**Algorithm 1** CFG-MP

---

**Require:** Velocity network $v_\theta$, initial noise $x_0$, guidance $w$, total steps $N$, projection iterations $K$, condition $y$.
1: **for** $i = 0$ **to** $N-1$ **do**
2:    $t_i \leftarrow i/N, \quad t_{i+1} \leftarrow (i+1)/N$
3:    {*1. Sampling Phase*}
4:    $v_i \leftarrow v^{\text{cfg}}_\theta(t_i, x_i, y)$
5:    $x_{i+1/2} \leftarrow x_i + 1/N \cdot v_i$
6:    {*2. Manifold Projection Phase*}
7:    $z_0 \leftarrow x_{i+1/2}$
8:    **for** $k = 1$ **to** $K$ **do**
9:       $z_k \leftarrow G(z_{k-1}, t_{i+1})$
10:    **end for**
11:    $x_{i+1} \leftarrow z_K$
12: **end for**
13: **RETURN** $x_N$

---

### 3.4. CFG-MP+: Acceleration scheme for manifold projection

The manifold projection step in Algorithm 1 can be framed as a fixed-point iteration (FPI) of the operator $G(\cdot, t)$, defined by the update $z_{k+1} = G(z_k, t)$. To improve convergence efficiency, FPI can be significantly enhanced using Anderson Acceleration (AA) (Anderson, 1965; Saad, 2025). AA speeds up the convergence of fixed-point schemes by constructing the next iterate as a linear extrapolation of past evaluations, with weights optimized to minimize the residual norm in a least-squares sense. We integrate AA into the manifold projection phase of Algorithm 1. Specifically, rather than utilizing the standard update $z_{k+1} = G(z_k, t)$, we compute the new iterate as a weighted combination of

historical states:

$$\begin{cases} f_k = G(z_k, t) - z_k \\ z_{k+1} = \sum_{i=k-m_k}^{k} \alpha_i^{k+1} (z_i + \beta f_i) \end{cases}$$

where $\beta$ denotes the damping factor and $m_k$ is the window size. The optimization weights $\boldsymbol{\alpha}^{k+1} := \{\alpha_j^{k+1}\}_{j=k-m_k}^{k}$ are determined by solving the constrained minimization problem:

$$\boldsymbol{\alpha}^{k+1} = \min_{\boldsymbol{\alpha}} \left\{ \left\| \sum_{i=k-m_k}^{k} \alpha_i f_i \right\| : \sum_{i=k-m_k}^{k} \alpha_i = 1 \right\}.$$

In practice, this is efficiently solved by reducing the constrained problem to an unconstrained least-squares system, requiring only a small matrix inversion that adds negligible computational overhead. This approach enhances both stability and convergence speed without requiring additional evaluations of the operator $G$. The complete procedure for CFG sampling, incorporating both manifold projection and Anderson Acceleration, is summarized in Algorithm 2 and is hereafter referred to as CFG-MP+.

---

**Algorithm 2** CFG-MP+ with AA(m,$\beta$)

**Require:** Velocity network $v_\theta$, initial noise $x_0$, guidance $w$, total steps $N$, window size $m$, damping factor $\beta$, projection iterations $K$, condition $y$.
1: **for** $i = 0$ **to** $N - 1$ **do**
2:     $t_i \leftarrow i/N, \quad t_{i+1} \leftarrow (i+1)/N$
3:     {*1. Sampling Phase*}
4:     $v_i \leftarrow v_\theta^{\text{cfg}}(t_i, x_i, y), \; x_{i+1/2} \leftarrow x_i + 1/N \cdot v_i$
5:     {*2. Manifold Projection Phase with AA(m,$\beta$)*}
6:     $z_0 \leftarrow x_{i+1/2}, \; z_1 \leftarrow G(z_0, t_{i+1}), \; f_0 \leftarrow z_1 - z_0$
7:     **for** $k = 1$ **to** $K - 1$ **do**
8:         $f_k \leftarrow G(z_k, t_{i+1}) - z_k, \quad m_k \leftarrow \min(k, m)$
9:         $\mathbf{F} \leftarrow [f_{k-m_k}, \dots, f_k], \quad \mathbf{Z} \leftarrow [z_{k-m_k}, \dots, z_k]$
        {*Construct matrices*}
10:        $\boldsymbol{\alpha}^{k+1} \leftarrow \arg\min_{\mathbf{1}^T \boldsymbol{\alpha}=1} \mathbf{F}\boldsymbol{\alpha}$    {*Solve least squares*}
11:       $z_{k+1} \leftarrow (\mathbf{Z} + \beta\mathbf{F})\boldsymbol{\alpha}^{k+1}$        {*AA mixing step*}
12:     **end for**
13:     $x_{i+1} \leftarrow z_K$
14: **end for**
15: **RETURN** $x_N$

---

# 4. Experimental Results

## 4.1. Class-to-image generation.

**Experimental setup.** We conduct experiments using the DiT-XL-2-256 model (Peebles & Xie, 2023) on the ImageNet $256 \times 256$ dataset (Deng et al., 2009) for class-conditional image generation. Performance is quantitatively evaluated using the Fréchet Inception Distance (FID) (Heusel et al., 2017) and Inception Score (IS) (Salimans

et al., 2016). Here, we utilize the Diff2Flow (D2F) technique (Schusterbauer et al., 2025) to convert the $\epsilon$-prediction model into a $v$-prediction formulation, enabling the application of our method. Then, two vanilla CFG baselines are established: one employing the DDIM sampler (Song et al., 2021) and the other utilizing the Euler method within the flow matching framework (Lipman et al., 2024). We further compare our approaches against several CFG variants within the DDIM framework, including Z-sampling (Bai et al., 2025), Re-sampling (Lugmayr et al., 2022), and FSG (Wang et al., 2025). For our proposed CFG-MP+, we fix the hyperparameters $(m, \beta) = (1, 1)$. We evaluate all methods across guidance scales $\omega \in \{1.5, 2.0, 2.5\}$ and number of function evaluations (NFEs) $\in \{60, 120\}$. All metrics are calculated based on 50,000 generated images (50 images per class), with results summarized in Table 1.

In terms of IS scores, our proposed CFG-MP and CFG-MP+ (hereafter referred to as CFG-MP/MP+) significantly outperform methods within the DDIM framework. Specifically, under the configuration of $\omega = 2.0$ and NFEs = 120, FSG achieves an IS gain of 5.50 over vanilla CFG(DDIM), whereas CFG-MP+ yields a markedly higher improvement of 14.78 relative to the CFG(D2F) baseline. Regarding the FID score, our methods effectively mitigate the sensitivity of FID to variations in guidance scales compared to other methods. These results underscore the superior robustness of CFG-MP/MP+, as they substantially mitigate the necessity for manual guidance scale tuning. Qualitative results are presented in Figure 2, where each row represents a distinct ImageNet category generated from identical Gaussian noise (NFEs $= 60, \omega = 2.0$). CFG-MP/MP+ consistently surpass the baselines across multiple visual dimensions: they achieve superior sharpness and preserve intricate textural details (e.g., the skin of the frog); they exhibit higher contrast and more vivid color saturation (e.g., the sea turtle and lion); and they effectively eliminate the blurry textures and over-smoothed surfaces prevalent in competing methods (e.g., the hamster).

CFG-DDIM    CFG-D2F    Z-samp.    Re-samp.    FSG    CFG-MP    CFG-MP+

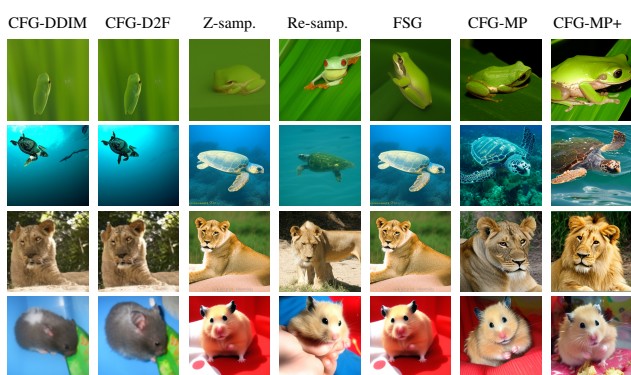

*Figure 2.* Qualitative comparison of generated samples on DiT-XL-2-256.

*Table 1.* FID (↓) and IS (↑) comparison on ImageNet256 (50k images) across different NFEs and guidance scales $\omega$. The best performance is set in **bold**, and the second best is set in underline.

| Guidance Scale | $\omega = 1.5$ | | | | $\omega = 2.0$ | | | | $\omega = 2.5$ | | | |
|---|---|---|---|---|---|---|---|---|---|---|---|---|
| NFEs | 60 | 120 | 60 | 120 | 60 | 120 | 60 | 120 | 60 | 120 | 60 | 120 |
| Methods | FID (↓) | | IS (↑) | | FID (↓) | | IS (↑) | | FID (↓) | | IS (↑) | |
| CFG(DDIM) | 11.76 | 9.61 | 61.39 | 61.86 | 13.29 | 12.17 | 67.98 | 69.25 | 15.83 | 14.71 | 72.53 | 73.17 |
| Z-sampling | 11.53 | 9.21 | 63.56 | 63.91 | 12.85 | 11.68 | 70.47 | 72.52 | 15.21 | 14.07 | 74.82 | 75.86 |
| Re-sampling | 11.48 | 8.56 | 66.98 | 67.87 | 12.92 | 11.63 | 71.88 | 73.07 | 15.17 | 14.14 | 75.25 | 76.01 |
| FSG | 11.42 | 9.14 | 67.20 | 65.73 | 12.76 | 11.50 | 72.31 | 74.75 | 15.04 | 14.01 | 75.98 | 77.14 |
| CFG(D2F) | 8.95 | **7.45** | 65.90 | 66.27 | 10.26 | **9.18** | 70.57 | 71.48 | 12.85 | 11.49 | 76.26 | 77.83 |
| CFG-MP | **8.85** | 7.77 | 72.85 | 73.11 | **10.03** | 9.26 | 78.74 | 79.25 | 12.51 | **11.48** | 82.45 | 83.74 |
| CFG-MP+ | 9.21 | 7.91 | **76.78** | **78.26** | 10.28 | 9.34 | **84.67** | **86.26** | 12.30 | 11.49 | **88.72** | **90.45** |

## 4.2. Text-to-image generation.

**Experimental setup.** We leverage SD3.5 (Esser et al., 2024) and Flux-dev (BlackForestLabs et al., 2025) as our backbone text-to-image generation models. To ensure a robust evaluation, we adopt prompts from diverse datasets, including DrawBench (Saharia et al., 2022), Pick-a-Pic (Kirstain et al., 2023), and GenEval (Ghosh et al., 2023). Performance is comprehensively assessed via a suite of state-of-the-art metrics, including CLIP score (Hessel et al., 2021), ImageReward (IR) score (Xu et al., 2023), PickScore (Kirstain et al., 2023), and HPSv2 score (Wu et al., 2023). Additionally, we evaluate our methods using the object-focused framework GenEval (Ghosh et al., 2023) for fine-grained analysis. For the choices of baselines, we compare CFG-MP and CFG-MP+ against prominent methods, including CFG-0* (Fan et al., 2025) and Rectified-CFG++ (Saini et al., 2025). Notably, Flux-dev is a guidance-distilled model, and our approaches adapt seamlessly to this model (Meng et al., 2023; BlackForestLabs et al., 2025) (see Appendix C.1 for details). For our proposed CFG-MP+, we consistently employ the AA(1,1) configuration.

The quantitative results for SD3.5 and Flux-dev are summarized in Table 2. Our methods consistently outperform the baselines across multiple metrics and prompt datasets under various guidance scales and NFEs. In particular, images generated by CFG-MP/MP+ exhibit significant improvements in HPSv2 and IR scores, indicating greater alignment with human preferences. Furthermore, GenEval results in Table 3 confirm that CFG-MP/MP+ excel in fine-grained compositional tasks, specifically in counting, color attribution, and spatial positioning, suggesting superior semantic alignment. Qualitative comparisons on SD3.5 are presented in Figure 3. CFG-MP/MP+ significantly surpasses the baselines in attribute binding (e.g., accurate counting and color mapping in Row 1), structural coherence (e.g., physically realistic frames in Rows 2-3), and contextual richness (e.g., fine-grained details in complex scenes in Row 4). Notably, our methods effectively mitigate common baseline failures such as attribute leakage and distorted geometries.

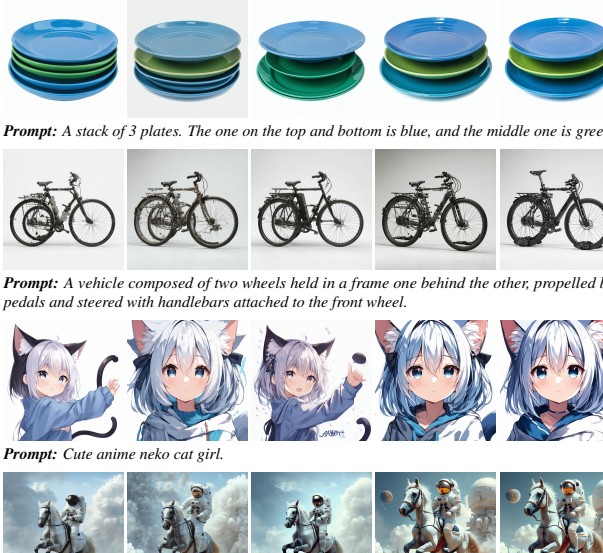

*Prompt: A stack of 3 plates. The one on the top and bottom is blue, and the middle one is green.*

*Prompt: A vehicle composed of two wheels held in a frame one behind the other, propelled by pedals and steered with handlebars attached to the front wheel.*

*Prompt: Cute anime neko cat girl.*

*Prompt: An astronaut riding a horse.*

*Figure 3.* Visual comparison on text-to-image generation tasks on SD3.5.

## 4.3. Principled analysis of the acceleration scheme.

Here, we discuss the effect of the acceleration scheme via the relative change defined by

$$r := \frac{\|v_\theta(t, z_k, y) - v_\theta(t, z_k, \emptyset)\| - \|v_\theta(t, z_0, y) - v_\theta(t, z_0, \emptyset)\|}{\|v_\theta(t, z_0, y) - v_\theta(t, z_0, \emptyset)\|}.$$

Consequently, a negative value ($r < 0$) signifies enhanced convergence via the acceleration scheme, whereas a positive value ($r > 0$) indicates performance degradation or potential divergence in the baseline FPI. Then, we compare the performance of vanilla FPI and AA(1,1)-accelerated FPI on operators $G(x, t)$ and $H(x, t)$, respectively, throughout the sampling process (with $K = 2, 4, 6$ iterations at each sampling step). This experiment is conducted on the DiT-XL-2-256 model.

Based on Figure 4, several key observations can be made.

*Table 2.* Quantitative Results for Flux-dev (left) and SD3.5 (right) across different NFEs and Guidance Scales. ↑ denotes that higher is better. The best performance is set in **bold**, and the second best is set in underline.

| | | Flux-dev (Guidance $\omega = 3$) | | | | | | | | | | SD3.5 (Different Guidance Scales) | | | | | | | |
| | | DrawBench | | | | Pick-a-Pic | | | | | | DrawBench | | | | Pick-a-Pic | | | |
| NFE | Method | Pick↑ | CLIP↑ | HPS↑ | IR↑ | Pick↑ | CLIP↑ | HPS↑ | IR↑ | $\omega$ | NFE | Pick↑ | CLIP↑ | HPS↑ | IR↑ | Pick↑ | CLIP↑ | HPS↑ | IR↑ |
|---|---|---|---|---|---|---|---|---|---|---|---|---|---|---|---|---|---|---|---|
| 60 | CFG | 22.91 | 33.28 | 28.85 | 0.93 | 22.19 | 32.32 | 29.87 | 0.97 | | 60 | 23.12 | 33.37 | 28.48 | 0.88 | 22.32 | 32.55 | 29.69 | 0.99 |
| | R-CFG++ | 23.09 | 33.48 | 28.89 | 0.94 | 22.21 | 32.34 | 29.89 | 0.99 | | | 23.15 | 33.39 | 28.61 | 0.90 | 22.35 | 32.60 | 29.89 | 1.02 |
| | CFG-0* | 23.12 | 33.39 | 28.87 | 0.93 | 22.20 | 32.34 | 29.90 | 0.91 | | | 23.13 | 33.40 | 28.54 | 0.87 | 22.34 | 32.57 | 29.76 | 1.00 |
| | CFG-MP | 23.01 | 33.37 | 29.50 | 0.95 | **22.24** | 32.36 | 29.91 | 0.96 | | | 23.18 | 33.41 | 28.76 | 0.93 | 22.37 | 32.60 | 29.77 | 1.03 |
| | CFG-MP+ | **23.23** | **33.49** | **30.50** | **1.02** | **22.24** | **32.39** | **29.92** | **1.05** | 3.0 | | **23.21** | **33.45** | **29.06** | **0.95** | **22.47** | **32.68** | **30.78** | **1.11** |
| 80 | CFG | 22.93 | 33.30 | 29.03 | 0.96 | 22.21 | 32.38 | 30.27 | 0.97 | | 100 | 23.13 | 33.37 | 28.50 | 0.89 | 22.34 | 32.57 | 29.73 | 1.00 |
| | R-CFG++ | 23.06 | **33.49** | 29.08 | 0.97 | 22.22 | 32.39 | 30.30 | 1.00 | | | 23.15 | 33.38 | 28.52 | 0.90 | 22.37 | 32.60 | 29.78 | 1.04 |
| | CFG-0* | 23.13 | 33.40 | 29.04 | 0.98 | 22.22 | 32.38 | 30.28 | 1.02 | | | 23.18 | 33.34 | 28.51 | 0.88 | 22.36 | 32.61 | 29.81 | 1.02 |
| | CFG-MP | 23.07 | 33.39 | 29.77 | 1.00 | **22.25** | **32.47** | 30.39 | 1.05 | | | 23.18 | 33.39 | 28.83 | 0.94 | 22.40 | 32.64 | 30.15 | 1.04 |
| | CFG-MP+ | **23.26** | 33.47 | **30.61** | **1.02** | **22.25** | 32.45 | **30.44** | **1.07** | | | **23.26** | **33.47** | **29.16** | **0.98** | **22.51** | **32.75** | **30.83** | **1.13** |
| 100 | CFG | 22.96 | 33.34 | 29.12 | 1.01 | 22.23 | 32.46 | 30.62 | 0.99 | | 60 | 23.06 | 33.30 | 28.89 | 0.88 | 22.26 | 32.50 | 30.18 | 1.04 |
| | R-CFG++ | 23.10 | **33.50** | 29.19 | 1.03 | 22.25 | **32.51** | 30.67 | 1.01 | | | **23.12** | 33.31 | 28.91 | 0.89 | **22.32** | 32.52 | 30.45 | 1.08 |
| | CFG-0* | 23.15 | 33.48 | 29.17 | 1.02 | 22.24 | **32.51** | 30.66 | 1.02 | | | 23.08 | 33.33 | 28.94 | 0.90 | 22.27 | 32.51 | 30.21 | 1.05 |
| | CFG-MP | 23.16 | 33.43 | 30.53 | **1.06** | 22.22 | 32.49 | 30.88 | 1.02 | | | 23.05 | 33.36 | 29.09 | **0.92** | 22.27 | 32.54 | 30.69 | 1.08 |
| | CFG-MP+ | **23.25** | **33.50** | **30.75** | 1.03 | **22.28** | 32.49 | **31.15** | **1.07** | 4.0 | | **23.12** | **33.38** | **29.29** | **0.92** | 22.29 | **32.55** | **30.94** | **1.16** |
| 120 | CFG | 22.98 | 33.36 | 29.30 | 1.02 | 22.28 | 32.51 | 30.90 | 1.00 | | 100 | 23.11 | 33.34 | 28.92 | 0.90 | 22.27 | 32.51 | 30.22 | 1.05 |
| | R-CFG++ | 23.12 | **33.52** | 29.57 | 1.03 | 22.29 | 32.52 | 30.94 | 1.03 | | | 23.14 | 33.36 | 28.95 | **0.91** | **22.33** | 32.53 | 30.25 | 1.08 |
| | CFG-0* | 23.16 | 33.49 | 29.60 | 1.03 | 22.30 | 32.53 | 30.95 | 1.04 | | | 23.12 | 33.35 | 28.96 | **0.91** | 22.31 | 32.54 | 30.24 | 1.07 |
| | CFG-MP | 23.17 | 33.46 | 30.18 | 1.03 | **22.35** | **32.57** | 31.01 | 1.05 | | | **23.15** | **33.40** | 28.96 | 0.90 | 22.31 | 32.54 | 30.78 | 1.11 |
| | CFG-MP+ | **23.26** | 33.50 | **30.79** | **1.07** | 22.34 | 32.55 | **31.21** | **1.10** | | | 23.14 | **33.40** | **29.35** | **0.91** | 22.32 | **32.56** | **31.15** | **1.15** |

*Table 3.* Quantitative results on GenEval with SD3.5 (cfg=4, NFE=60). All values are in percentages (%) and ↑. The best performance is set in **bold**, and the second best is set in underline.

| Method | Single | Two Obj. | Count | Colors | Pos. | Attr. | Overall |
|---|---|---|---|---|---|---|---|
| CFG | 90.00 | 72.73 | 52.50 | 81.91 | 13.00 | 40.00 | 58.36 |
| CFG-0* | 91.25 | 74.75 | 53.75 | 77.66 | 14.00 | 44.00 | 59.24 |
| R-CFG++ | 91.25 | **75.76** | 52.50 | 80.85 | 15.00 | 42.00 | 59.57 |
| CFG-MP | **91.50** | 74.75 | **56.75** | 80.85 | 16.00 | **45.00** | 60.81 |
| CFG-MP+ | **91.50** | **75.76** | 56.25 | **82.98** | **17.00** | **45.00** | **61.41** |

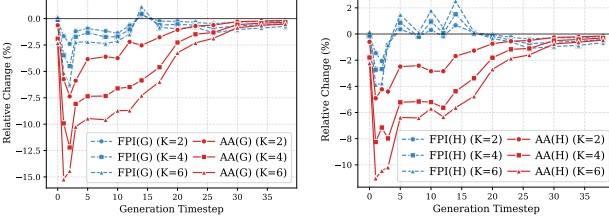

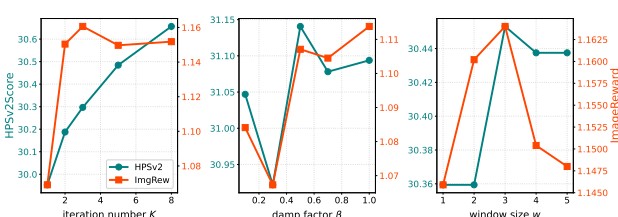

*Figure 4.* The improved performance of operator $G$ (left) and $H$ (right) under AA(1,1) with $w = 1.5$, $K = 2, 4, 6$.

### 4.4. Ablation study.

First, the vanilla $G(x,t)$ operator consistently outperforms the vanilla $H(x,t)$; for instance, when $K = 6$ and the inference step is in $[1, 2]$, the relative change $r$ for FPI($G$) ranges from -6% to -5%, outperforming FPI($H$), which hovers between -4% and -2%. Notably, the vanilla iterations for both operators may occasionally exhibit divergence, particularly when the inference step is in $[5, 15]$: FPI($G$) diverged once, whereas FPI($H$) diverged at least three times. These instances of instability underscore the necessity of employing AA to improve convergence stability on non-contractive operators, as established in Pollock & Rebholz (2021). Ultimately, the AA(1,1)-accelerated versions consistently surpass their vanilla counterparts across various $K$ values and sampling stages, demonstrating the robust effectiveness of the AA scheme in accelerating the decay of the prediction gap.

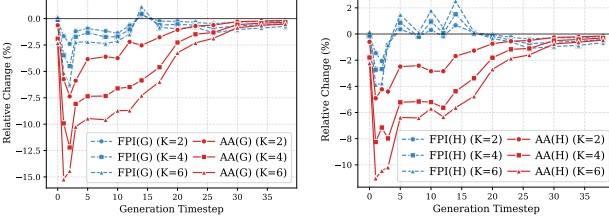

*Figure 5.* Ablation studies on the Anderson Acceleration hyperparameters.

We mainly conduct ablation studies on the following key hyperparameters of our AA-based method:
**The number of FPIs under AA.** We vary the number of AA(1) iterations among $\{1, 2, 3, 5, 8\}$. Our study shows that

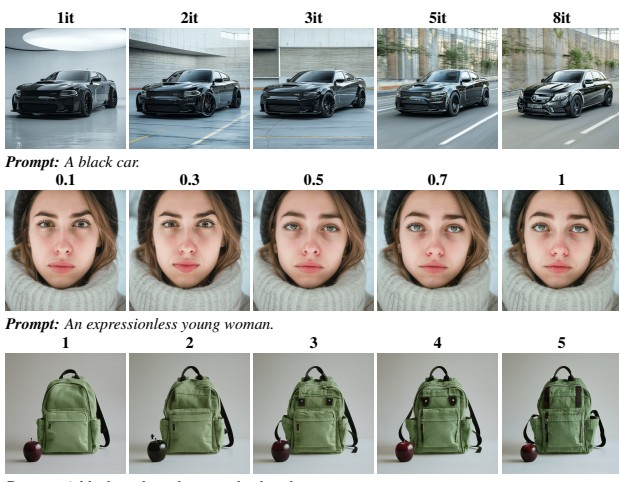

**Prompt:** *A black car.*

**Prompt:** *An expressionless young woman.*

**Prompt:** *A black apple and a green backpack.*

*Figure 6.* Qualitative ablation study on different hyperparameters: FPI iterations (top), damping factors (middle), and window size (bottom).

*Table 4.* Sensitivity analysis of AA hyperparameters in terms of relative change $-r$ (%) ($\uparrow$) on DiT-XL-2-256.

| $m$ | $\beta$ | $K=3$ | $K=4$ | $K=5$ |
|---|---|---|---|---|
|   | 0.2 | 9.31 | 13.52 | 16.85 |
|   | 0.4 | 9.32 | 13.54 | 16.82 |
| 1 | 0.6 | 9.37 | 13.64 | 16.76 |
|   | 0.8 | 9.36 | 13.56 | 16.74 |
|   | 1.0 | **9.44** | **13.76** | **16.97** |
|   | 0.2 | 9.30 | 13.49 | 16.70 |
|   | 0.4 | 9.35 | 13.52 | 16.72 |
| 2 | 0.6 | 9.33 | 13.54 | 16.75 |
|   | 0.8 | 9.31 | 13.53 | 16.73 |
|   | 1.0 | 9.32 | 13.54 | 16.82 |
|   | 0.2 | 9.30 | 13.50 | 16.71 |
|   | 0.4 | 9.29 | 13.51 | 16.73 |
| 3 | 0.6 | 9.31 | 13.52 | 16.74 |
|   | 0.8 | 9.28 | 13.51 | 16.72 |
|   | 1.0 | 9.31 | 13.59 | 16.80 |

increasing the iterations at each inference step improves the HPSv2 score, while the IR score stabilizes once the iteration number reaches 2. Given that the computational cost scales with the number of FPIs, we recommend setting FPI $= 2$ in practice as an optimal trade-off between computational efficiency and generation quality.

**The sensitivity of AA hyperparameters in terms of relative change.** We further evaluate the sensitivity of AA hyperparameters using the relative change $r$ defined in Section 4.3. In Table 4, we report $-r$ under different AA window sizes $m$, damping factors $\beta$, and FPI iterations $K \in \{3, 4, 5\}$, where a larger value indicates a stronger reduction of the prediction gap. The results are stable across a broad range of $(m, \beta)$ choices. In particular, the simple setting $m = 1, \beta = 1.0$ consistently achieves the best relative reduction for all tested $K$, supporting our default choice AA(1,1) in the main experiments in Section 4.1 and 4.2.

**The choice of damping factor for Damped AA.** We vary the AA damping factor among $\{0.1, 0.3, 0.5, 0.7, 1\}$. Empirical results suggest that a damping factor $\beta \in \{0.5, 1.0\}$ consistently yields peak performance for the operator $G(x, t)$.

**The choice of window size for AA.** We evaluate the AA window size across $\{1, 2, 3, 4, 5\}$. While a window size of 3 yields the best performance for the operator $G(x, t)$ in terms of HPSv2 and IR scores, a larger window size entails higher computational overhead (more FPIs). Therefore, we adopt a window size of 1 in practice for efficiency.

We also give some qualitative results, as shown in Figure 6. Our CFG-MP+ is robust to different choices of these three AA hyperparameters.

## 5. Conclusions

In this work, we provide an optimization-based interpretation of classifier-free guidance, clarifying its objective, limitations, and sensitivity to the guidance scale in flow-based generative models. By reformulating CFG sampling as a constrained homotopy optimization problem, we introduced CFG-MP, an iterative manifold-projection scheme that mitigates the prediction gap between conditional and unconditional predictions during inference. Combined with Anderson Acceleration, the resulting CFG-MP+ improves sampling stability, convergence, and generation quality in a training-free manner across class-to-image and text-to-image generation tasks.

Although CFG-MP and CFG-MP+ effectively reduce the prediction gap during sampling, this benefit is obtained through additional fixed-point iterations and thus requires potentially more NFEs. This extra inference-time computation is useful for improving sample quality and robustness, but it may limit practical deployment when cost is critical. A promising direction is to move prediction-gap mitigation from sampling to training, for example, by treating the prediction gap as a reinforcement-learning reward or training-time regularization signal. Such a strategy could encourage the model to produce better-aligned conditional and unconditional predictions before inference, thereby reducing the need for additional computation during generation. Another natural direction is to adapt our manifold-projection formulation and Anderson-accelerated fixed-point scheme to diffusion models. Since DDIM and related diffusion samplers also follow structured sampling dynamics, extending CFG-MP/MP+ to these samplers may also reduce guidance scale sensitivity.

## Acknowledgements

This work was partially supported by the National Natural Science Foundation of China under Grant 12571564, Guangdong Basic and Applied Research Foundation 2024A1515012347, Hong Kong Research Grant Council (HKRGC) GRF grants 16307325, 16306124, and 16307023, Interdisciplinary Research Program of HUST 2024JCYJ005, and National Key Research and Development Program of China 2023YFC3804500.

## Impact Statement

This paper presents work whose goal is to advance the field of Machine Learning. There are many potential societal consequences of our work, none which we feel must be specifcally highlighted here.

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

## A. Related Works

**Classifier-Free Guidance and its improved variants.** Classifier-Free Guidance (CFG) (Ho & Salimans, 2021) has become the standard approach for steering text-to-image generative models. Despite its success, standard CFG often suffers from structural distortions and over-saturation. To address these issues, several recent works have proposed refinements across different generative frameworks. For flow models, Saini et al. (2025) analyzes the off-manifold drift in rectified flow models and introduces a scheduled interpolation to maintain sample faithfulness while enhancing prompt alignment. Similarly, Fan et al. (2025) identifies that inaccurate velocity estimations in the early training stages of flow matching models lead to suboptimal trajectories, proposing an optimized scale and a zero-initialization strategy to improve generation quality. Beyond flow-based models, advancements in diffusion sampling have also gained significant attention. Wang et al. (2025) reformulates the guidance process as a fixed-point iteration problem and introduces a prediction gap perspective, utilizing longer-interval subproblems to improve prompt alignment. In a parallel effort to enhance semantic consistency, Bai et al. (2025) adopts a self-reflection mechanism that incorporates inversion steps during the denoising process to dynamically refine the guidance direction. These methods collectively push the boundaries of CFG by addressing its theoretical and practical limitations. Echoing the perspective of Wang et al. (2025), our work also addresses the refinement of CFG through the lens of the prediction gap. However, while prior studies have primarily focused on the empirical utility of closing this gap, we provide, to the best of our knowledge, the first rigorous theoretical analysis characterizing the fundamental relationship between sampling fidelity and the prediction gap. This theoretical grounding allows us to move beyond heuristic adjustments, providing a principled justification to minimize the gap and preserve the integrity of the sampling trajectory.

**Homotopy Optimization.** Homotopy optimization methods (Watson & Haftka, 1989; Lin et al., 2023), also known as continuation methods, provide a robust framework for solving non-convex optimization problems by tracing a continuous path of solutions from a simplified auxiliary problem to the target objective. In the context of generative modeling, the sampling process can be elegantly reinterpreted through this lens: the noise-to-data transition effectively forms a homotopy path where the objective function is gradually annealed over time. The iterative trajectory of generative models, such as diffusion models and flow matching models, indeed evolves from a simple prior to a complex data distribution, and thus exhibits a striking structural parallel to the path-tracking mechanism in homotopy optimization. Recognizing this intrinsic similarity motivates us to investigate the theoretical connections between these two domains. By formally viewing the sampling steps through the lens of homotopy optimization, we can treat sampling not merely as probabilistic updates, but as a formal homotopy optimization problem. This perspective establishes a rigorous foundation for incorporating advanced numerical techniques to improve the fidelity of the generative trajectory.

**Anderson Acceleration.** To enhance the convergence efficiency of iterative solvers, Anderson Acceleration (AA) (Anderson, 1965; Saad, 2025) has emerged as a powerful technique for accelerating fixed-point iterations. Originally developed for self-consistent field calculations, AA estimates the fixed point by maintaining a history of previous iterates and computing an optimal linear combination that minimizes the residual norm within a local window. In contrast to standard Picard iteration, which only utilizes the most recent information, AA exploits multi-step historical information to accelerate convergence toward the fixed point. The convergence behavior of AA has been a subject of intensive scholarly investigation in recent years. For a rigorous treatment of the theoretical results in this domain, we refer the readers to Toth & Kelley (2015); Pollock & Rebholz (2021); Wei et al. (2023).

## B. Proofs

### B.1. Proof of Theorem 3.1

*Proof.* We begin by introducing some foundational notation. In the context of CFM, let $p_t$ denote the density function of the random variable $x_t = tx_1^y + (1-t)x_0$. For a fixed $x_1$, the conditional probability density of $x_t$ is defined as:

$$p_t(x|x_1) = \frac{1}{(2\pi(1-t)^2)^{d/2}} \exp\left(-\frac{\|x - tx_1\|^2}{2(1-t)^2}\right).$$

The marginal density can then be expressed as:

$$p_t^y(x) = \int_{\mathbb{R}^d} p_t(x|x_1) p_1^y(x_1) dx_1$$

$$= \int_{\mathbb{R}^d} \frac{1}{(2\pi(1-t)^2)^{d/2}} \exp\left(-\frac{\|x-tx_1\|^2}{2(1-t)^2}\right) p_1^y(x_1)dx_1.$$

Next, we define the posterior data distribution $p_1^y(x|z,t)$ for any $z \in \mathbb{R}^d$ via Bayes' formula:

$$p_1^y(x|z,t) = \frac{p_t(z|x)p_1^y(x)}{p_t^y(z)} \tag{B.1}$$

$$= \frac{\exp\left(-\frac{\|z-tx\|^2}{2(1-t)^2}\right) p_1^y(x)}{\int_{\mathbb{R}^d} \exp\left(-\frac{\|z-tx_1\|^2}{2(1-t)^2}\right) p_1^y(x_1)dx_1}. \tag{B.2}$$

Similarly, for UFM, the posterior data distribution $p_1^\emptyset(x|z,t)$ is defined as:

$$p_1^\emptyset(x|z,t) = \frac{\exp\left(-\frac{\|z-tx\|^2}{2(1-t)^2}\right) p_1^\emptyset(x)}{\int_{\mathbb{R}^d} \exp\left(-\frac{\|z-tx_1\|^2}{2(1-t)^2}\right) p_1^\emptyset(x_1)dx_1}. \tag{B.3}$$

We now proceed to prove Theorem 3.1. We first consider the case of $v_{t,\emptyset}^*(x)$. Let $x_t^\emptyset = x$ for simplicity. The explicit form of the loss functional $\mathcal{L}^\emptyset(f)$ is given by:

$$\mathcal{L}^\emptyset(f) = \mathbb{E}_{t,x_0,x_1}\left[\|f(t,(1-t)x_0+tx_1) - (x_1-x_0)\|^2\right]$$

$$= \int_0^1 \int_{\mathbb{R}^d} \int_{\mathbb{R}^d} \|f(t,x) - (x_1-x_0)\|^2 \frac{1}{(2\pi)^{d/2}} \exp\left(-\frac{\|x_0\|^2}{2}\right) p_1^\emptyset(x_1)dx_0dx_1dt$$

$$= \int_0^1 \frac{1}{(2\pi(1-t)^2)^{d/2}} \int_{\mathbb{R}^d} \int_{\mathbb{R}^d} \left\|f(t,x) - \left(x_1 - \frac{x-tx_1}{1-t}\right)\right\|^2 \exp\left(-\frac{\|x-tx_1\|^2}{2(1-t)^2}\right) p_1^\emptyset(x_1)dxdx_1dt$$

$$= \int_0^1 \int_{\mathbb{R}^d} \int_{\mathbb{R}^d} \left\|f(t,x) - \frac{x_1-x}{1-t}\right\|^2 p_t(x|x_1)p_1^\emptyset(x_1)dx_1dxdt.$$

To find the minimizer of this functional, we optimize it pointwise. For any fixed $x \in \mathbb{R}^d$, we consider the function:

$$K_t(z) = \int_{\mathbb{R}^d} \left\|z - \frac{x_1-x}{1-t}\right\|^2 p_t(x|x_1)p_1^\emptyset(x_1)dx_1, \quad z \in \mathbb{R}^d.$$

Since $K_t(z)$ is strictly convex with respect to $z$, it possesses a unique minimizer. Setting the gradient to zero, we have:

$$\nabla_z K_t(z) = 2 \int_{\mathbb{R}^d} \left(z - \frac{x_1-x}{1-t}\right) p_t(x|x_1)p_1^\emptyset(x_1)dx_1 = 0,$$

which yields the unique minimizer:

$$z^* = \frac{\int_{\mathbb{R}^d} \frac{x_1-x}{1-t} p_t(x|x_1)p_1^\emptyset(x_1)dx_1}{\int_{\mathbb{R}^d} p_t(x|x_1)p_1^\emptyset(x_1)dx_1} = \frac{\mathbb{E}_{y\sim p_1^\emptyset(\cdot|x,t)}[y] - x}{1-t} = v_{t,\emptyset}^*(x).$$

Consequently, for any $f \in C((0,1)\times\mathbb{R}^d, \mathbb{R}^d)$, it holds that:

$$\mathcal{L}^\emptyset(f) = \int_0^1 \int_{\mathbb{R}^d} K_t(f(t,x))dxdt$$

$$\geq \int_0^1 \int_{\mathbb{R}^d} K_t(v_{t,\emptyset}^*(x))dxdt$$

$$= \int_0^1 \int_{\mathbb{R}^d} \int_{\mathbb{R}^d} \left\|v_{t,\emptyset}^*(x) - \frac{x_1-x}{1-t}\right\|^2 p_t(x|x_1)p_1^\emptyset(x_1)dx_1dxdt$$

$$= \mathcal{L}^\emptyset(v_{t,\emptyset}^*).$$

The proof for $v_{t,y}^*(x)$ follows by replacing $p_1^\emptyset(x_1)$ with $p_1^y(x_1)$ throughout the derivation. This completes the proof. $\quad\square$

## B.2. Proof of Theorem 3.3

*Proof.* We again focus on the unconditional case $v_{t,\emptyset}^*(x)$. According to Theorem 3.1, the optimal velocity field can be expressed as:

$$(1-t)v_{t,\emptyset}^*(x) = \mathbb{E}_{y \sim p_1^\emptyset(\cdot|x,t)}[y] - x$$
$$= \frac{\int_{\mathbb{R}^d} x_1 p_t(x|x_1) p_1^\emptyset(x_1) dx_1}{\int_{\mathbb{R}^d} p_t(x|x_1) p_1^\emptyset(x_1) dx_1} - x.$$

In parallel:

$$\frac{1}{2}\text{dist}_{t\mathcal{K}_\emptyset}^2(x, 1-t) - \frac{1-t}{2}\|x\|^2 = -(1-t)^2 \log\left\{\int_{\mathbb{R}^d} \exp\left(-\frac{\|tx_1 - x\|^2}{2(1-t)^2}\right) p_1^\emptyset(x_1) dx_1\right\} - \frac{1-t}{2}\|x\|^2.$$

By taking the gradient with respect to $x$, we obtain:

$$-\nabla_x\left\{\frac{1}{2}\text{dist}_{t\mathcal{K}_\emptyset}^2(x, 1-t) - \frac{1-t}{2}\|x\|^2\right\}$$

$$= (1-t)^2 \frac{\int_{\mathbb{R}^d} \nabla_x \exp\left(-\frac{\|tx_1 - x\|^2}{2(1-t)^2}\right) p_1^\emptyset(x_1) dx_1}{\int_{\mathbb{R}^d} \exp\left(-\frac{\|tx_1 - x\|^2}{2(1-t)^2}\right) p_1^\emptyset(x_1) dx_1} + (1-t)x$$

$$= (1-t)^2 \frac{\int_{\mathbb{R}^d} \frac{tx_1 - x}{(1-t)^2} \exp\left(-\frac{\|tx_1 - x\|^2}{2(1-t)^2}\right) p_1^\emptyset(x_1) dx_1}{\int_{\mathbb{R}^d} \exp\left(-\frac{\|tx_1 - x\|^2}{2(1-t)^2}\right) p_1^\emptyset(x_1) dx_1} + (1-t)x.$$

Simplifying the above using the definition of $p_t(x|x_1)$, we have:

$$-\nabla_x\left\{\frac{1}{2}\text{dist}_{t\mathcal{K}_\emptyset}^2(x, 1-t) - \frac{1-t}{2}\|x\|^2\right\} = \frac{\int_{\mathbb{R}^d} tx_1 p_t(x|x_1) p_1^\emptyset(x_1) dx_1}{\int_{\mathbb{R}^d} p_t(x|x_1) p_1^\emptyset(x_1) dx_1} - x + (1-t)x$$

$$= t\left(\frac{\int_{\mathbb{R}^d} x_1 p_t(x|x_1) p_1^\emptyset(x_1) dx_1}{\int_{\mathbb{R}^d} p_t(x|x_1) p_1^\emptyset(x_1) dx_1} - x\right)$$

$$= t(1-t)v_{t,\emptyset}^*(x).$$

The conditional case for $v_{t,y}^*(x)$ follows the identical derivation by substituting $p_1^\emptyset(x_1)$ with $p_1^y(x_1)$ throughout. This completes the proof. □

## B.3. Proof of Theorem 3.4

*Proof.* The argument is based on a simple orthogonal decomposition. We have:

$$\|v_{t,y}^*(x) - v_\theta^{\text{cfg}}(t,x,y)\|^2 = \|v_{t,y}^*(x) - v_\theta^{\text{cfg}*}(t,x,y) + v_\theta^{\text{cfg}*}(t,x,y) - v_\theta^{\text{cfg}}(t,x,y)\|^2$$

$$= \|v_{t,y}^*(x) - v_\theta^{\text{cfg}*}(t,x,y)\|^2 + \|v_\theta^{\text{cfg}*}(t,x,y) - v_\theta^{\text{cfg}}(t,x,y)\|^2$$

$$+ 2\langle v_{t,y}^*(x) - v_\theta^{\text{cfg}*}(t,x,y), v_\theta^{\text{cfg}*}(t,x,y) - v_\theta^{\text{cfg}}(t,x,y)\rangle$$

$$= \|v_{t,y}^*(x) - v_\theta^{\text{cfg}*}(t,x,y)\|^2 + \|v_\theta^{\text{cfg}*}(t,x,y) - v_\theta^{\text{cfg}}(t,x,y)\|^2$$

$$+ 2(w^* - w)\langle v_{t,y}^*(x) - v_\theta^{\text{cfg}*}(t,x,y), v_\theta(t,x,y) - v_\theta(t,x,\emptyset)\rangle.$$

If we define the function $h(w) := \|v_{t,y}^*(x) - v_\theta(t,x,\emptyset) - w(v_\theta(t,x,y) - v_\theta(t,x,\emptyset))\|^2$, then the optimal $w^*$ satisfies $\frac{d}{dw}h(w^*) = 0$. Let $v_\theta^{\text{cfg}*}(t,x,y) := v_\theta(t,x,\emptyset) + w^*(v_\theta(t,x,y) - v_\theta(t,x,\emptyset))$, we have:

$$\langle v_\theta(t,x,y) - v_\theta(t,x,\emptyset), v_{t,y}^*(x) - v_\theta^{\text{cfg}*}(t,x,y)\rangle = 0,$$

then we just plug this relationship in and we have:

$$\|v_{t,y}^*(x) - v_\theta^{\text{cfg}}(t,x,y)\|^2 = \|v_{t,y}^*(x) - v_\theta^{\text{cfg}*}(t,x,y)\|^2 + \|v_\theta^{\text{cfg}*}(t,x,y) - v_\theta^{\text{cfg}}(t,x,y)\|^2$$

$$= \|v_{t,y}^*(x) - v_\theta^{\text{cfg}*}(t,x,y)\|^2 + (w^* - w)^2\|v_\theta(t,x,y) - v_\theta(t,x,\emptyset)\|^2.$$

Thus we have finished the proof. □

## B.4. Basics of Anderson Acceleration

We consider the fixed-point problem of finding $x^* \in \mathbb{R}^n$ such that $x = F(x)$, where $F : \mathbb{R}^n \to \mathbb{R}^n$ is a nonlinear mapping. We define the stage-$k$ differences between iterates and their corresponding residuals as:

$$e_k := x_k - x_{k-1}, \quad r_k := F(x_{k-1}) - x_{k-1}.$$

The Anderson Acceleration (AA) algorithm with depth $m \geq 0$ (where $m = 0$ recovers the ordinary Picard Iteration) and damping factors $0 < \beta_k \leq 1$ proceeds according to following general algorithm:

- **Step 0:** Choose $x_0 \in \mathbb{R}^n$.

- **Step 1:** Find $r_1 = F(x_0) - x_0$. Set $x_1 = x_0 + r_1$.

- **Step $k+1$:** For $k = 1, 2, 3, \ldots$, set $m_k = \min\{k, m\}$.

  a. Find the current residual $r_{k+1} = F(x_k) - x_k$.
  b. Solve the minimization problem for the coefficients $\boldsymbol{\alpha}^{k+1} = \{\alpha_j^{k+1}\}_{j=k-m_k}^k$:

  $$\min_{\sum_{j=k-m_k}^k \alpha_j^{k+1}=1} \left\| \sum_{j=k-m_k}^k \alpha_j^{k+1} r_{j+1} \right\|.$$

  c. For a given $\beta_k$, update the next iterate:

  $$x_{k+1} = \sum_{j=k-m_k}^k \alpha_j^{k+1} x_j + \beta_k \sum_{j=k-m_k}^k \alpha_j^{k+1} r_{j+1} = (1-\beta_k) \sum_{j=k-m_k}^k \alpha_j^{k+1} x_j + \beta_k \sum_{j=k-m_k}^k \alpha_j^{k+1} F(x_j).$$

For more details of Anderson Acceleration, see Anderson (1965); Pollock & Rebholz (2021); Saad (2025).

## C. Experiment Details

### C.1. Flux-dev and the Guidance Distillation Mechanism

According to BlackForestLabs et al. (2025); Meng et al. (2023), the Flux-dev model makes use of the guidance distillation technique:

$$v_\theta^{\text{flux}}(x, t, y, w) \approx v_\theta(x, t, \emptyset) + w \cdot (v_\theta(x, t, y) - v_\theta(x, t, \emptyset)),$$

where $w \geq 1$ is the guidance scale. This distillation technique reduces the number of model evaluations in each inference step, since the original CFG uses the network twice, whereas here we only need one calculation to evaluate $v_\theta^{\text{flux}}(x, t, y, w)$ by providing four inputs.

We observe that

$$v_\theta^{\text{flux}}(x, t, y, w) = \nabla_x(f_t^{\emptyset}(x) + w(f_t^y(x) - f_t^{\emptyset}(x))), \quad v_\theta^{\text{flux}}(x, t, y, 0) = \nabla_x f_t^{\emptyset}(x),$$
$$v_\theta^{\text{flux}}(x, t, y, w) - v_\theta^{\text{flux}}(x, t, y, 0) = w \cdot \nabla_x(f_t^y(x) - f_t^{\emptyset}(x)),$$

where we use the notation in (3.4) and (3.6). Here we assume that $v_\theta^{\text{flux}}(x, t, y, w) = v_\theta(x, t, \emptyset) + w \cdot (v_\theta(x, t, y) - v_\theta(x, t, \emptyset))$, which means the guidance distillation performs well during the training stage.

Now we define the function

$$F_t'(x) := \tfrac{w}{2t(1-t)}(f_t^y(x) - f_t^{\emptyset}(x)) \tag{C.1}$$

$$= \tfrac{1}{2t(1-t)}\left( f_t^{\emptyset}(x) + w(f_t^y(x) - f_t^{\emptyset}(x)) \right) - \tfrac{1}{2t(1-t)} f_t^{\emptyset}(x). \tag{C.2}$$

So for the $\mathcal{M}_t$ constraint we again have

$$\mathcal{M}_t = \{x|0 = \nabla F'_t(x) = \nabla \tfrac{w}{2t(1-t)}(f^y_t(x) - f^\emptyset_t(x)) = w\nabla F_t(x)\}$$
$$= \{x|v^{\text{flux}}_\theta(x,t,y,w) = v^{\text{flux}}_\theta(x,t,y,0)\}$$

Here, $F_t(x)$ is defined in Section 3.3. As a result, projecting a point onto $\mathcal{M}_t$ is equivalent to minimizing $F'_t(x)$. Now we again use the incremental gradient descent scheme for $F'_t(x)$. Specifically, we take a gradient step on $-\tfrac{1}{2t(1-t)}f^\emptyset_t(x)$ and then take a gradient step on $\tfrac{1}{2t(1-t)}\left(f^\emptyset_t(x) + w(f^y_t(x) - f^\emptyset_t(x))\right)$ (which follows the same form as (3.7) and (3.7)), and we define the operator $G'(x,t)$ as:

$$G'(x,t) := x - \tfrac{1}{2}\Delta t \cdot v^{\text{flux}}_\theta(x,t,y,0) + \tfrac{1}{2}\Delta t \cdot v^{\text{flux}}_\theta(t, x - \tfrac{1}{2}\Delta t \cdot v^{\text{flux}}_\theta(x,t,y,0), y, w). \tag{C.3}$$

We now replace $G$ with $G'$ in the manifold projection phase of Algorithm 1 and Algorithm 2 (here, the $w$ in (C.3) is the guidance scale used in the CFG sampling phase), and we get the CFG-MP and CFG-MP+ sampling methods when using Flux 1-dev as our backbone model.

## C.2. More experimental results

### C.2.1. VALIDATION OF THE OPTIMAL CHOICE $a = b = \tfrac{1}{2}\Delta t$ WHEN CONSTRUCTING $G(x,t)$

We justify the optimality of $a = b = \tfrac{1}{2}\Delta t$ when constructing operator $G(x,t)$. Specifically, we can set $a = b = \lambda \in (0,1)$ and we have:

$$G_\lambda(x,t) := x - \lambda\Delta t \cdot v_\theta(t,x,\emptyset) + \lambda\Delta t \cdot v_\theta(t, x - \lambda\Delta t \cdot v_\theta(t,x,\emptyset), y),$$

and we recover the setting in CFG-MP/MP+ in Section 3.3 by setting $\lambda = 0.5$. We now give results of CFG-MP+ with different values of $\lambda$ ($0.1, 0.3, 0.5, 0.7, 0.9$). We first give results of FID and IS with DiT-XL-2-256 model (see table 5), and then we give results of HPSv2 and IR scores with SD3.5 model (see table 6).

*Table 5.* Ablation study of CFG-MP+ with varying $\lambda$ on FID and IS metrics.

| Metric | NFEs | $\lambda = 0.1$ | $\lambda = 0.3$ | $\lambda = 0.5$ | $\lambda = 0.7$ | $\lambda = 0.9$ |
|---|---|---|---|---|---|---|
| FID ($\downarrow$) | 60 | 9.35 | 9.31 | **9.21** | **9.21** | 9.23 |
| | 120 | 7.96 | 7.94 | 7.90 | **7.89** | 7.92 |
| IS ($\uparrow$) | 60 | 76.52 | 76.43 | **76.78** | 76.44 | 76.23 |
| | 120 | 77.92 | 78.24 | **78.53** | 78.51 | **78.53** |

*Table 6.* Ablation study of CFG-MP+ with varying $\lambda$ on HPSv2 and IR metrics.

| Metric | Steps | $\lambda = 0.1$ | $\lambda = 0.3$ | $\lambda = 0.5$ | $\lambda = 0.7$ | $\lambda = 0.9$ |
|---|---|---|---|---|---|---|
| HPSv2 ($\uparrow$) | 20 | 30.97 | 31.04 | **31.18** | 31.14 | 31.15 |
| | 30 | 31.01 | 31.06 | 31.21 | 31.20 | **31.22** |
| IR ($\uparrow$) | 20 | 1.10 | 1.11 | **1.12** | 1.04 | 1.03 |
| | 30 | 1.12 | **1.13** | 1.12 | 1.07 | 1.05 |

From these results, we observe that $\lambda = 0.5$ shows superior performance in terms of different metrics and different models. So we consistently choose $\lambda = 0.5$ for all tasks in our experiment section.

### C.2.2. GENERATION TIME COMPARISON

We further compare the wall-clock generation time on Stable Diffusion 3.5 to verify that the proposed projection iterations do not introduce a disproportionate computational burden. As shown in Table 7, all methods are competing under the same NFE settings in terms of per-image generation time. Meanwhile, CFG-MP/MP+ consistently improve human-preference metrics. For example, at NFE $= 60$, CFG-MP+ improves IR over the strong R-CFG++ baseline by 7.4%, while maintaining a comparable generation time. This confirms that the gains of CFG-MP/MP+ are not obtained from an unfair wall-clock-time advantage, but from more effective sampling dynamics.

*Table 7.* Performance and efficiency comparison across different sampling methods on Stable Diffusion 3.5. Time denotes wall-clock generation time per image in seconds.

| NFE | Metric | CFG | R-CFG++ | CFG-0* | CFG-MP | CFG-MP+ |
|---|---|---|---|---|---|---|
| 60 | Time ($\downarrow$) | 15.43s | 15.22s | 15.67s | **14.98s** | 15.04s |
| | HPSv2 ($\uparrow$) | 30.18 | 30.45 | 30.21 | 30.69 | **30.94** |
| | IR ($\uparrow$) | 1.04 | 1.08 | 1.05 | 1.08 | **1.16** |
| 100 | Time ($\downarrow$) | 25.89s | 25.42s | 25.27s | **25.08s** | 25.24s |
| | HPSv2 ($\uparrow$) | 30.22 | 30.25 | 30.24 | 30.78 | **31.15** |
| | IR ($\uparrow$) | 1.05 | 1.08 | 1.07 | 1.11 | **1.15** |

*Table 8.* Sensitivity analysis of Anderson Acceleration hyperparameters on Stable Diffusion 3.5 in terms of ImageReward (IR, $\uparrow$) and Human Preference Score (HPSv2, $\uparrow$).

| $m$ | $\beta$ | $K = 3$ | | $K = 4$ | | $K = 5$ | |
|---|---|---|---|---|---|---|---|
| | | IR | HPSv2 | IR | HPSv2 | IR | HPSv2 |
| 1 | 0.2 | 1.08 | 30.92 | 1.11 | 31.04 | 1.13 | 31.07 |
| | 0.4 | 1.09 | 30.90 | 1.12 | 31.03 | 1.14 | 31.06 |
| | 0.6 | 1.10 | 30.96 | 1.14 | 31.08 | 1.15 | 31.09 |
| | 0.8 | 1.10 | 30.93 | 1.13 | 31.05 | 1.14 | 31.08 |
| | 1.0 | **1.12** | **31.00** | **1.15** | **31.12** | **1.18** | **31.16** |
| 2 | 0.2 | 1.05 | 30.92 | 1.11 | 31.04 | 1.15 | 31.07 |
| | 0.4 | 1.06 | 30.90 | 1.11 | 31.02 | 1.14 | 31.05 |
| | 0.6 | 1.06 | 30.95 | 1.13 | 31.06 | 1.16 | 31.07 |
| | 0.8 | 1.07 | 30.94 | 1.12 | 31.07 | 1.15 | 31.11 |
| | 1.0 | 1.09 | 30.96 | 1.13 | 31.10 | 1.16 | 31.13 |
| 3 | 0.2 | 1.07 | 30.92 | 1.09 | 31.05 | 1.14 | 31.08 |
| | 0.4 | 1.07 | 30.91 | 1.10 | 31.07 | 1.13 | 31.12 |
| | 0.6 | 1.09 | 30.95 | 1.12 | 31.06 | 1.15 | 31.11 |
| | 0.8 | 1.08 | 30.94 | 1.11 | 31.05 | 1.14 | 31.09 |
| | 1.0 | 1.09 | 30.95 | 1.12 | 31.09 | 1.15 | 31.14 |

### C.2.3. SENSITIVITY OF ANDERSON ACCELERATION HYPERPARAMETERS

We further study the sensitivity of Anderson Acceleration on Stable Diffusion 3.5, focusing on the window size $m$ and damping factor $\beta$. Table 8 reports the IR and HPSv2 scores obtained by CFG-MP+ under different choices of $(m, \beta)$ and projection iterations $K$. The results are stable across a broad range of AA hyperparameters. Increasing $K$ consistently improves both IR and HPSv2, while the simple setting $m = 1, \beta = 1.0$ achieves the best performance for all tested $K$. This supports our default AA(1,1) choice and indicates that the acceleration scheme does not require delicate tuning in practice.

### C.2.4. DETAILS OF SECTION 4.1

The full FID and IS comparison across guidance scales is reported in Table 1. To further evaluate the performance and stability of different sampling methods under varying guidance intensities, we introduce two relative metrics: the *FID Growth Ratio* and the *Relative IS Gain*.

**FID Growth Ratio** ($R_{\Delta\text{FID}}$). Standard FID scores often degrade as the guidance scale $\omega$ increases; this is the phenomenon known as over-guidance. To quantify a method's robustness against this degradation, we define the FID growth ratio as:

$$R_{\Delta\text{FID}} = \frac{\text{FID}_{\text{method}}(\omega_{i+1}) - \text{FID}_{\text{method}}(\omega_i)}{\text{FID}_{\text{base}}(\omega_{i+1}) - \text{FID}_{\text{base}}(\omega_i)}$$

where $\omega_i$ and $\omega_{i+1}$ represent adjacent guidance scales. A lower $R_{\Delta\text{FID}}$ indicates superior stability, meaning the method is less sensitive to the distribution shifts induced by stronger guidance.

**Relative IS Gain** ($G_{\text{IS}}$). While guidance scale $\omega$ is typically increased to enhance semantic clarity (reflected by IS), it often comes at the cost of diversity. We measure the IS gain by calculating the relative percentage improvement in Inception Score

compared to the respective baseline at a fixed $\omega$:

$$G_{\text{IS}}(\omega) = \frac{\text{IS}_{\text{method}}(\omega) - \text{IS}_{\text{base}}(\omega)}{\text{IS}_{\text{base}}(\omega)} \times 100\%$$

This metric highlights the capacity of a sampler to extract stronger semantic features without requiring excessively high guidance scales, thereby potentially preserving better image quality.

To ensure a fair comparison across different generative paradigms, we employ framework-specific baselines. For methods operating within the DDIM framework (e.g., Z-sampling, Re-sampling, and FSG), we use the vanilla CFG(DDIM) as the baseline. Conversely, for flow-based methods, including our proposed CFG-MP and CFG-MP+, the state-of-the-art CFG(D2F) is utilized as the baseline.

Table 9. FID growth ratios ($\Delta\text{FID}_{\text{m}}/\Delta\text{FID}_{\text{b}}$).

| Methods | $\omega : 1.5 \to 2.0$ | | $\omega : 2.0 \to 2.5$ | |
|---|---|---|---|---|
| | 60 | 120 | 60 | 120 |
| Z-sampling | 0.863 | 0.965 | 0.929 | 0.941 |
| Re-sampling | 0.941 | 1.199 | 0.886 | 0.988 |
| FSG | 0.876 | 0.922 | 0.898 | 0.988 |
| CFG-MP | 0.901 | 0.861 | 0.958 | 0.961 |
| CFG-MP+ | **0.817** | **0.827** | **0.780** | **0.931** |

Table 10. Relative IS gain (%) vs. baselines.

| Methods | $\omega = 1.5$ | $\omega = 2.0$ | $\omega = 2.5$ |
|---|---|---|---|
| | 60 / 120 | 60 / 120 | 60 / 120 |
| Z-sampling | +3.5 / +3.3 | +3.7 / +4.7 | +3.2 / +3.7 |
| Re-sampling | +9.1 / +9.7 | +5.7 / +5.5 | +3.8 / +3.9 |
| FSG | +9.5 / +6.3 | +6.4 / +7.9 | +4.8 / +5.4 |
| CFG-MP | +10.6 / +10.3 | +11.6 / +10.9 | +8.1 / +7.6 |
| CFG-MP+ | **+16.5 / +18.1** | **+20.0 / +20.7** | **+16.3 / +16.2** |

Our experimental results demonstrate that CFG-MP/MP+ significantly enhance the robustness of FID and consistently improve IS across various guidance scales $\omega$. Quantitatively, both methods maintain $R_{\Delta\text{FID}}$ below 1.0 across all guidance intervals and NFE settings, indicating superior stability against distribution degradation compared to the framework-specific baselines. Notably, CFG-MP+ achieves its most robust performance in the $\omega : 2.0 \to 2.5$ interval at 60 NFEs with a ratio of 0.780, whereas all DDIM-based variants exhibit ratios exceeding 0.88. Regarding $G_{\text{IS}}(\omega)$, CFG-MP+ yields substantial relative IS gains ranging from +16.2% to +20.7% over the CFG(D2F) baseline. The peak relative improvement is observed at $\omega = 2.0$, reaching +20.7% at 120 NFEs, which again outperforms all DDIM-based variants.

### C.2.5. VALIDATION OF THE EFFECTIVENESS OF OPERATOR $G(x, t)$

We provide more results for Section 4.3. Specifically, we evaluate the prediction gap decay among different inference steps $(25, 30, 35, 40)$ (see Figure 7). These results empirically demonstrate that the forward-backward splitting-based operator $G(x, t)$ can reduce the prediction gap through its fixed-point iteration (or the AA(1,1) accelerated version), and the AA(1,1) scheme can significantly accelerate this process and stabilize the non-contractive operator $G(x, t)$ in the middle stage of sampling.

### C.3. Computational environment

All experiments are conducted on a 4-A100(40GB) GPU server.

### C.4. More results on image generation

We list some additional results to further demonstrate the effectiveness of our CFG-MP and CFG-MP+. Figure 8 is generated with DiT-XL-2-256 model, Figure 9, 10, and 11 are generated with Flux-dev model, and Figure 12, 13, and 14 are generated with SD3.5 model.

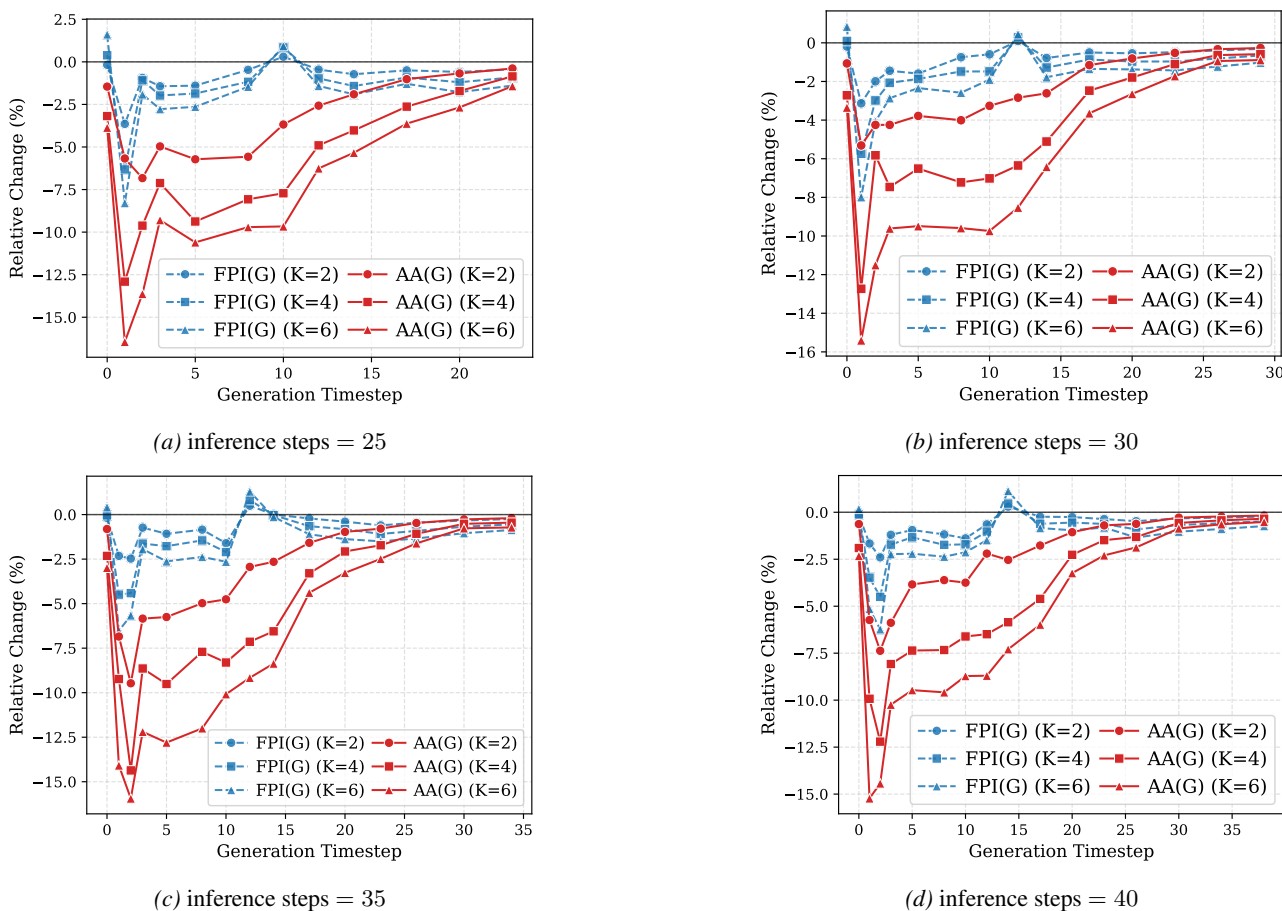

*Figure 7.* Comparison of the relative change across different inference steps (25,30,35,40).

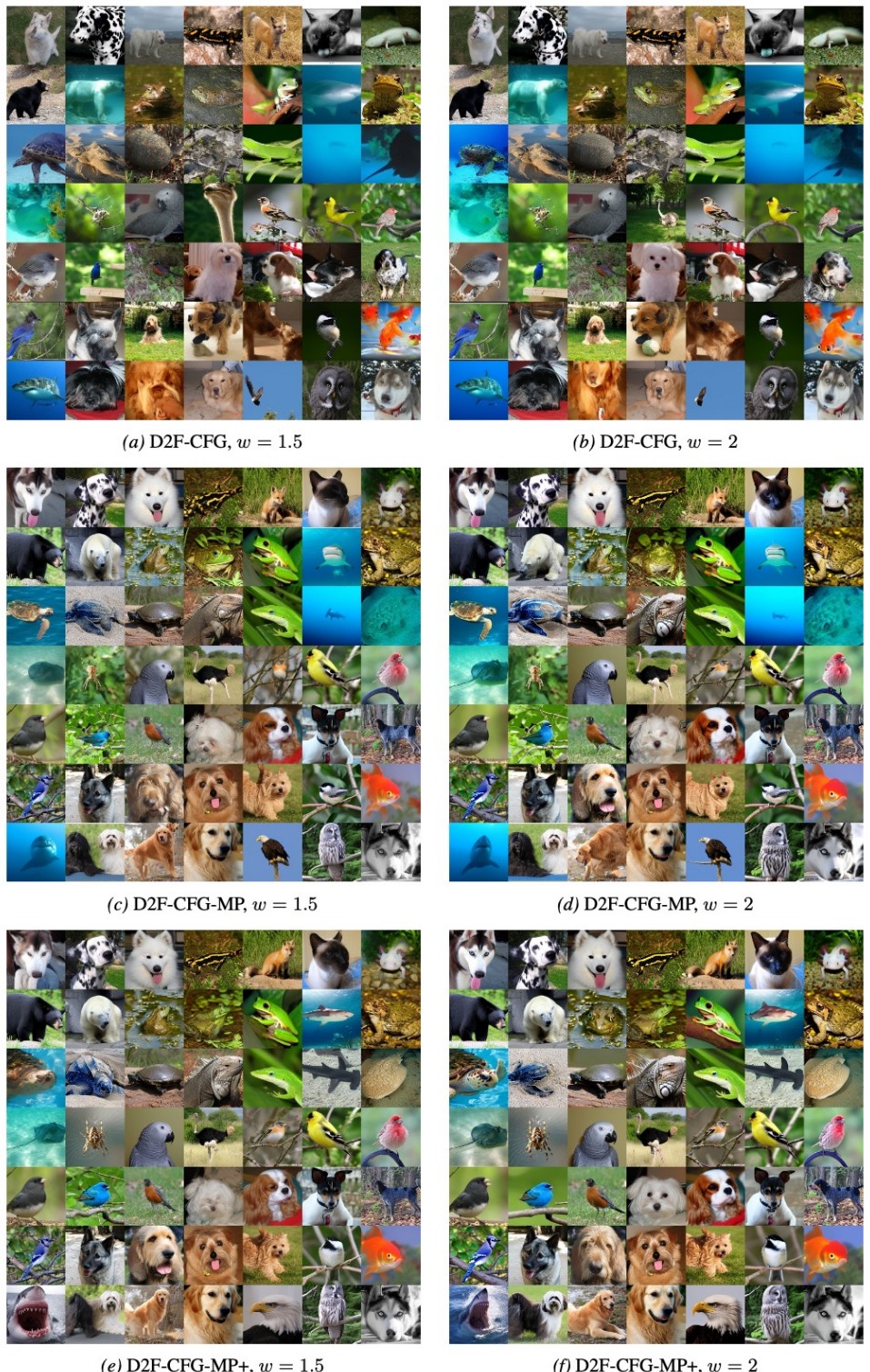

*(a)* D2F-CFG, $w = 1.5$

*(b)* D2F-CFG, $w = 2$

*(c)* D2F-CFG-MP, $w = 1.5$

*(d)* D2F-CFG-MP, $w = 2$

*(e)* D2F-CFG-MP+, $w = 1.5$

*(f)* D2F-CFG-MP+, $w = 2$

*Figure 8.* Class to image generation (ImageNet256, DiT-XL-2-256) with NFEs $= 60$.

**Prompt:** *Macro shot of an eye filled with extreme sorrow. The iris reflects a blurred candlelight,with a single, about-to-fall tearatthe corner, shot with a 105mmlens.*

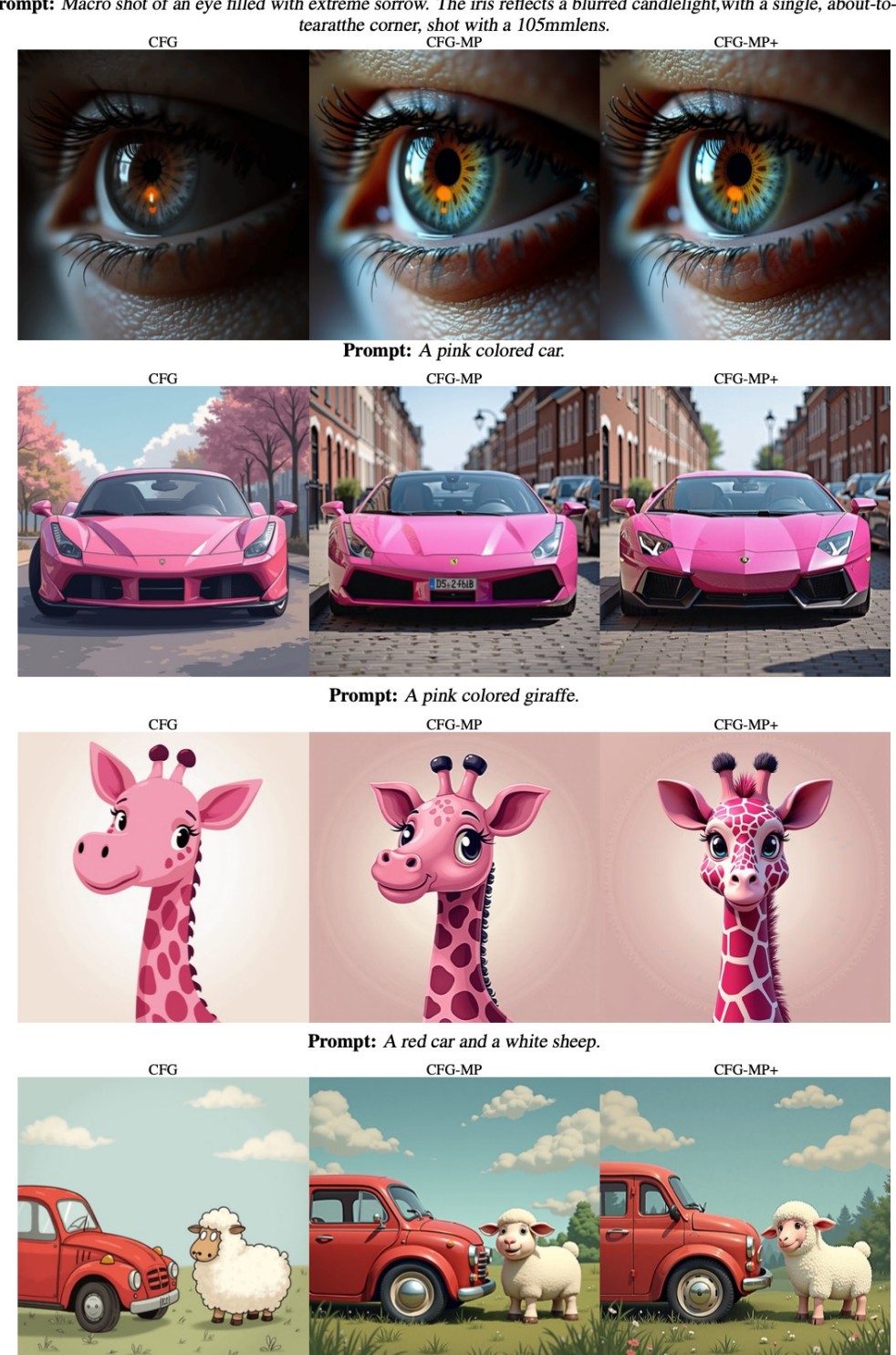

*Figure 9.* **Qualitative comparison of CFG variants on Flux-dev (Set 1).** Here we use $w = 3$ and NFEs $= 50$.

**Prompt:** *An elephant is behind a tree.*

CFG | CFG-MP | CFG-MP+

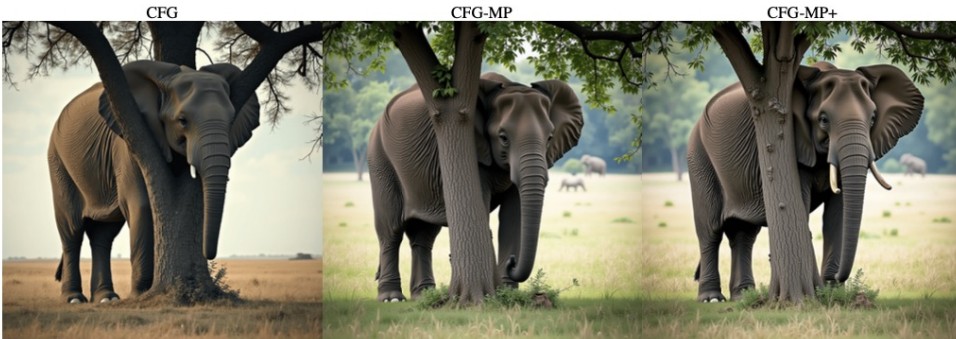

**Prompt:** *A brown bird and a blue bear.*

CFG | CFG-MP | CFG-MP+

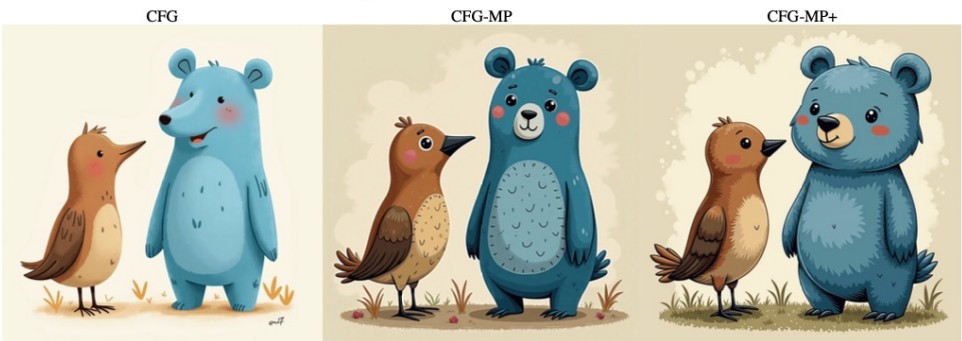

**Prompt:** *A connection point by which firefighters can tap into a water supply.*

CFG | CFG-MP | CFG-MP+

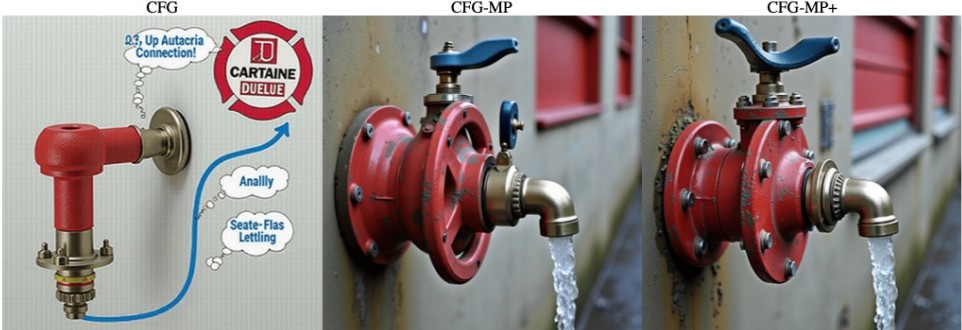

**Prompt:** *An appliance or compartment which is artificially kept cool and used to store food and drink.*

CFG | CFG-MP | CFG-MP+

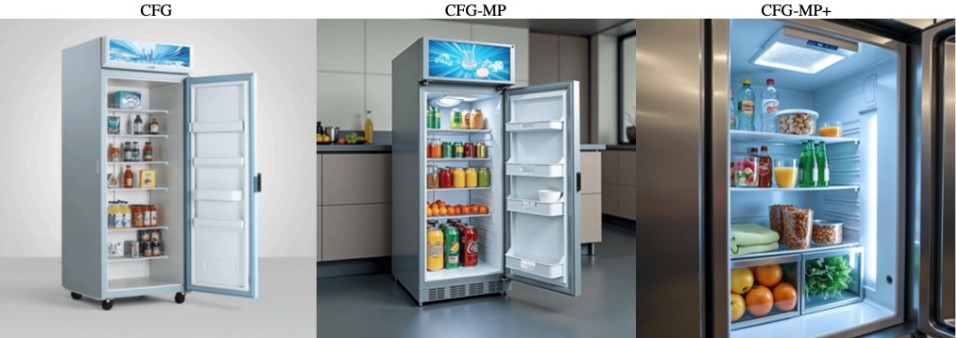

*Figure 10.* **Qualitative comparison of CFG variants on Flux-dev (Set 2).** Here we use $w = 3$ and NFEs $= 50$.

**Prompt:** *Cutaway diagram of stunning 3d rendered humanoid female cyborg flowerpunk mechanical puppet, unreal engine, octane render, highly detailed, volumetric light, ray tracing, knolling inside parts around.*

**Prompt:** *little bunny.*

**Prompt:** *birthday card with the exact text happy birthday user.*

**Prompt:** *a girl with long silver hair, she looks 15 old, wearing cute dress, anime-style.*

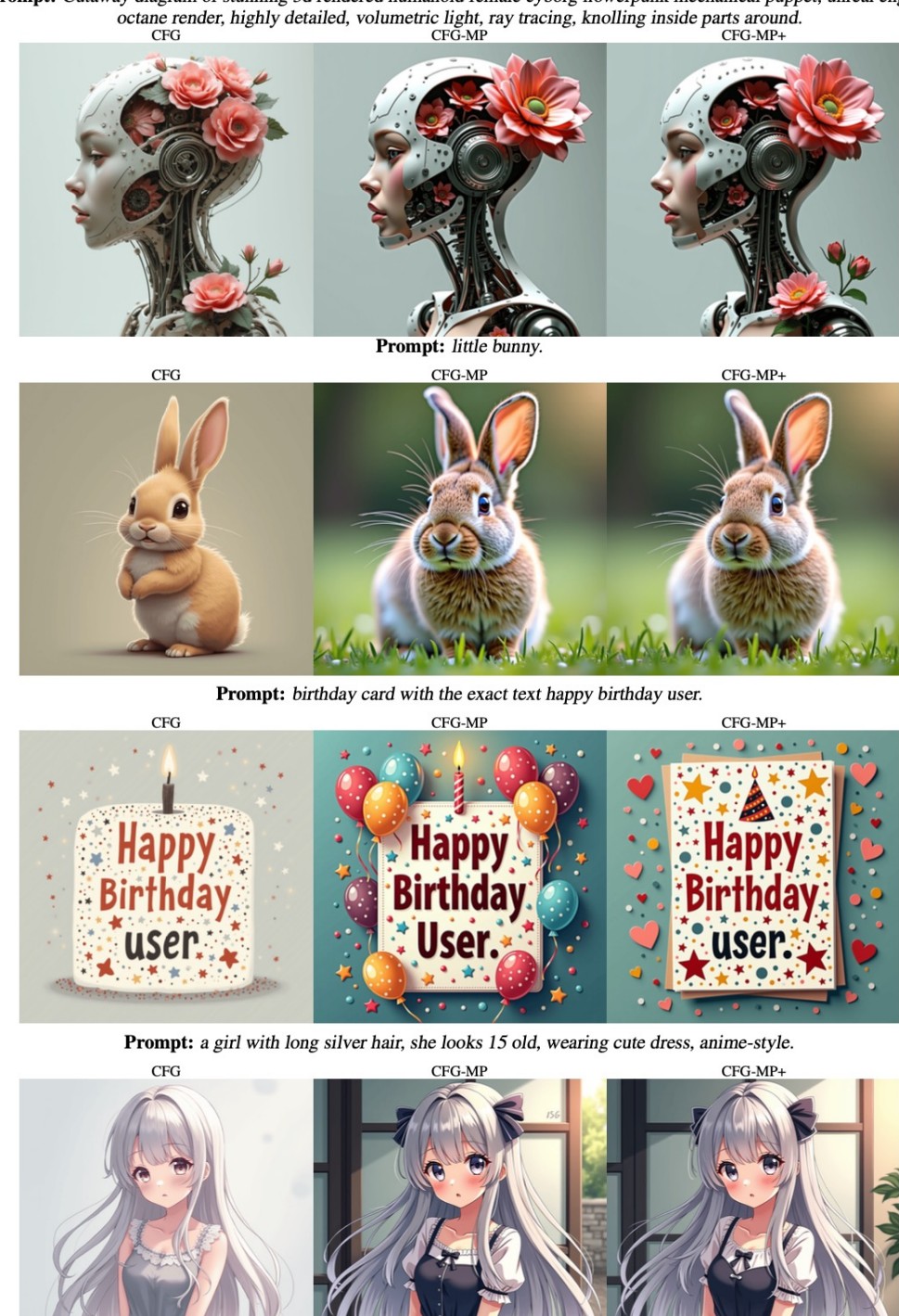

*Figure 11.* **Qualitative comparison of CFG variants on Flux-dev (Set 3).** Here we use $w = 3$ and NFEs $= 50$.

**Prompt:** *Close-up picture of a parrot dropping a spoon.*

CFG        CFG-MP        CFG-MP+

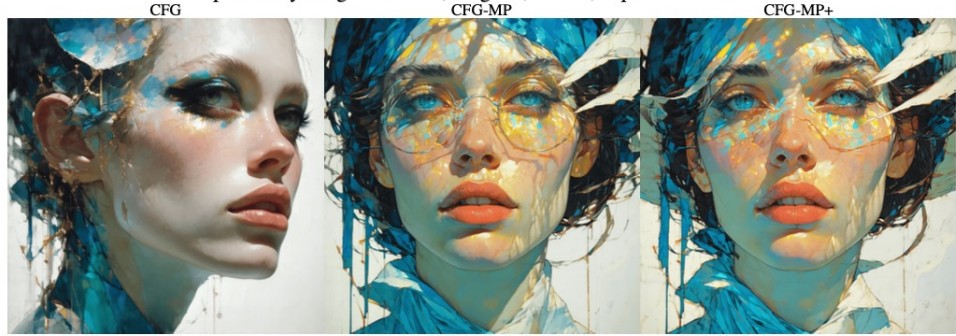

**Prompt:** *Cover, hyperdetailed painting, luminism, bar lighting, complex, head and shoulders portrait, 4k resolution concept art portrait by Greg Rutkowski, Artgerm, WLOP, Alphonse Mucha.*

CFG        CFG-MP        CFG-MP+

**Prompt:** *female sherlock holmes.*

CFG        CFG-MP        CFG-MP+

**Prompt:** *A giant eagle monster art.*

CFG        CFG-MP        CFG-MP+

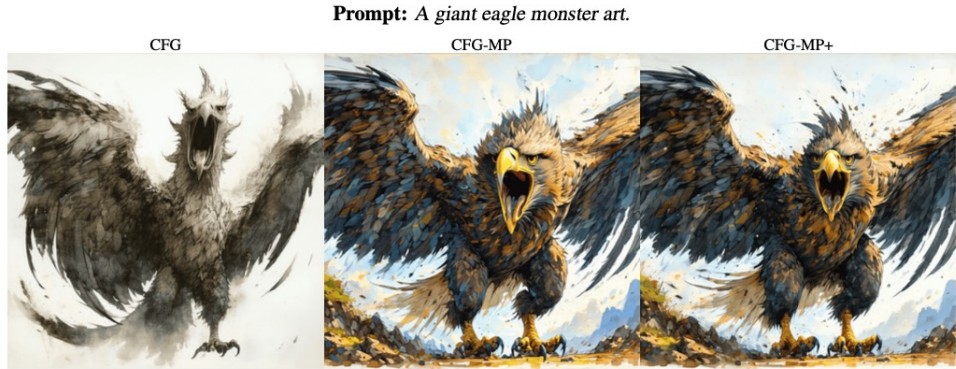

*Figure 12.* **Qualitative comparison of CFG variants on SD3.5 (Set 1).** Here we use $w = 3$ and NFEs $= 60$.

**Prompt:** *portrait of a cyborg e-girl painting.*

CFG           CFG-MP           CFG-MP+

**Prompt:** *a cat who fall in a swimmingpool.*

CFG           CFG-MP           CFG-MP+

**Prompt:** *cinematic still of an adorable walking robot in the desert, at sunset.*

CFG           CFG-MP           CFG-MP+

**Prompt:** *high quality portrait illustration of Dennis Wilson from The Beach Boys.*

CFG           CFG-MP           CFG-MP+

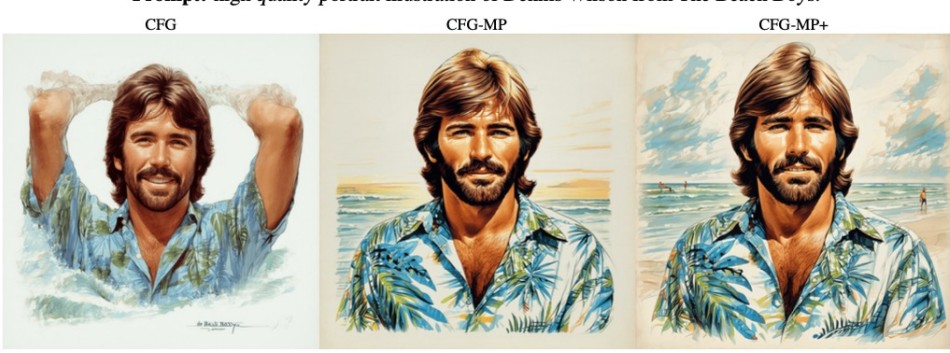

*Figure 13.* **Qualitative comparison of CFG variants on SD3.5 (Set 2).** Here we use $w = 3$ and NFEs $= 60$.

**Prompt:** *A little blonde girl busy eating a cup of yogurt.*

**Prompt:** *Realistic owl.*

**Prompt:** *A wall painting of a girl.*

**Prompt:** *A war weary hamster soldier.*

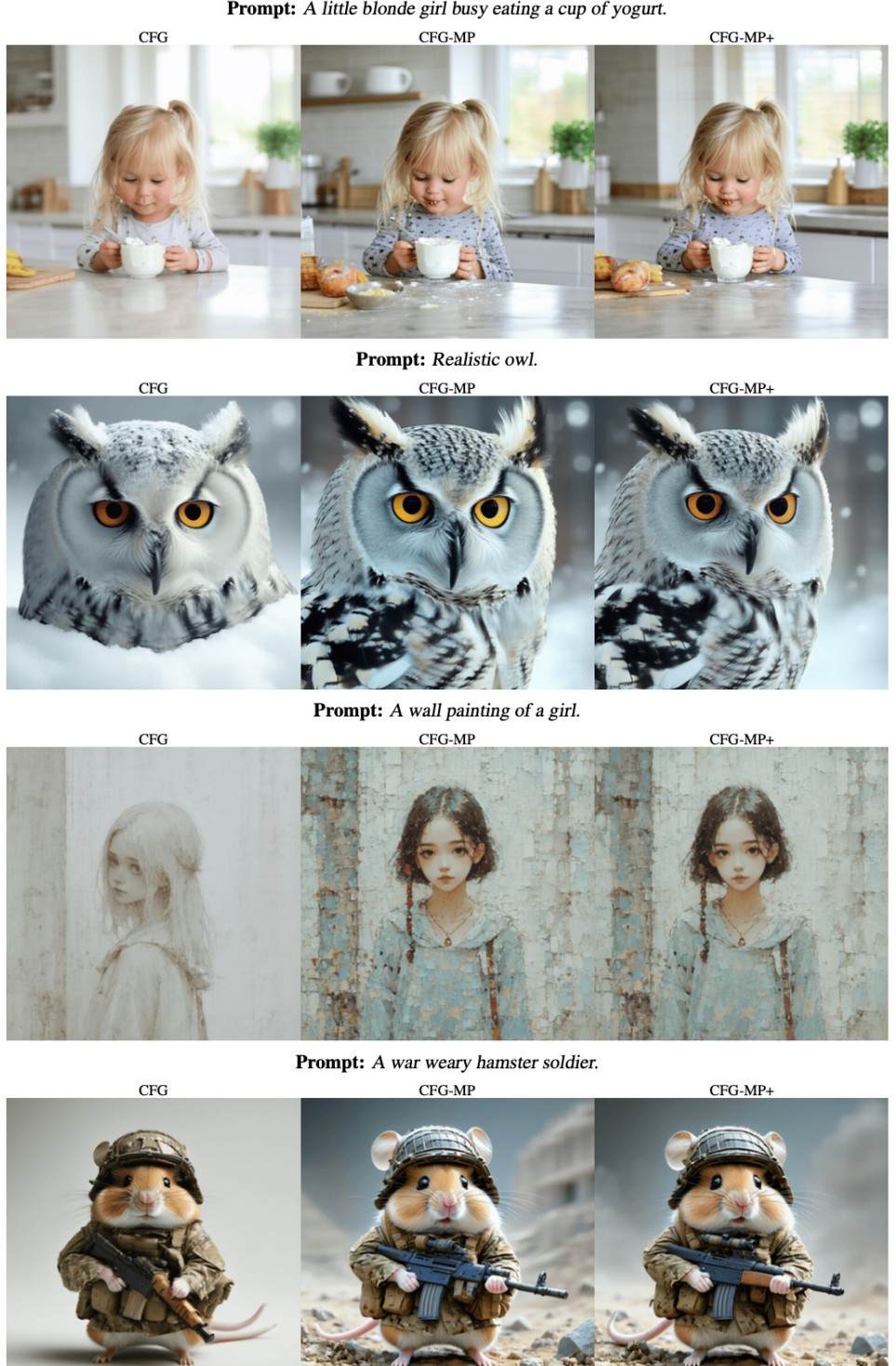

*Figure 14.* **Qualitative comparison of CFG variants on SD3.5 (Set 3).** Here we use $w = 3$ and NFEs $= 60$.

