# OpenReview forum: "Improving Classifier-Free Guidance of Flow Matching via Manifold Projection"
_ICML.cc/2026/Conference — ICML 2026 regular_

### Official Review · Reviewer_Lj88 · 2026-03-07

**Soundness:** 2
**Presentation:** 3
**Significance:** 3
**Originality:** 3
**Overall Recommendation:** 4
**Confidence:** 4

**Summary:**

This paper aims to reduce the research gap regarding the lack of theoretical interpretations of classifier-free guidance (CFG) when applied to flow-matching models. By orthogonally decomposing the approximation of the ideal velocity field, the authors argue that the approximation error can be reduced by minimizing the prediction gap. This prediction gap is minimized through fixed-point iteration using gradient descent, and the optimization is further enhanced by applying Anderson acceleration. Experimental results show that the proposed method improves the quality of generated images compared to previous CFG methods.

**Compliance With Llm Reviewing Policy:**

Affirmed.

**Final Justification:**

Since the authors' rebuttal fully addressed by questions, my final recommendation is raised from 3 to 4.

**Key Questions For Authors:**

- The paragraph right below the Eq. 3.7 explains that according to theorem 3.3, the two-step incremental update involving the terms $f^{\phi}$ and $f^y$ can be instead expressed with parameterized velocity fields $v_{\theta}$. However, Theorem 3.3 establishes the relationship between $f^{y}$ and the optimal velocity field $v^{\*}\_{y}$ . Why is it justified to replace the optimal $v\_{y}^{\*}$ with the optimization-in-progress $v\_{\theta}(y)$? Similarly, why is it valid to use $v\_{\theta}(\phi)$ instead of the optimal $v\_{\phi}^{\*}$?

- Replacing $v^*$ with $v_\theta$ appears to remove explicit dependence on the smoothing parameter $\sigma$ in the smoothed squared-distance function. According to Definition 3.2, the smoothed distance function deviates from the exact distance function when $\sigma$ is large. Does the removal of explicit dependence on $\sigma$ imply that its magnitude does not influence the method’s effectiveness?

- In the geometric interpretation in Section 3.3, the smoothed distance function is approximated by the exact distance function. However, Definition 3.2 indicates that the approximation is accurate only when the smoothing parameter approaches zero. Since $(1-t)$ here corresponds to the smoothing parameter, this suggests that at early timesteps of generation(i.e. when t is small), the smoothed distance would differ substantially from the exact distance. Is there an analysis of how this approximation behaves as the timestep evolves?

**Limitations:**

The authors did not discuss the limitations of their method.

**Strengths And Weaknesses:**

**Strengths**
- Motivations and contributions are clearly stated.
- The paper introduced a new perspective of viewing the CFG sampling as a homotopy optimization.
- The prediction gap reduction problem was addressed in a novel way compared to previous works.

**Weaknesses**
- The proposed method reduces the prediction gap by manifold-projected fixed-point iteration, but relies on insufficiently explained approximations on its procedure.
- The paper presents extensive evaluations across varying guidance scales and NFEs. However, the persuasiveness of the evaluation results is limited since the effective guidance scale and NFEs might differ depending on the sampling method used. Clarifying the optimal NFE and guidance scale combinations for each of the baseline methods for a fair comparison would strengthen the claim.

---

> ### Author Rebuttal · Authors · 2026-03-28
>
> **[W1\&Q1] approximation in manifold:**
> 1. To reduce guidance scale sensitivity, our method focuses on narrowing the prediction gap. The manifold $\mathcal{M}_t$ is an illustrative formalization; our approach remains effective even in its absence. Per Remark 3.6 and Fig. 4, the iterative minimization of $F_t(x)$ proceeds regardless of the manifold, proving it is not a prerequisite for performance.
>
> 2. The approximations in the derivation of our iteration scheme have two parts:
>
>  - We introduce a proxy manifold $\mathcal{M}'\_t := \lbrace z \mid v^* \_{ t,y }(z) = v^* \_{t,\emptyset }(z) \rbrace$ defined by the ideal velocity field $v^* $, rather than the learned manifold $\mathcal{M}\_{t} := \lbrace z \mid v\_{\theta}(t,z,y) = v\_{\theta}(t,z,\emptyset) \rbrace$. This distinction is necessary because an arbitrary vector field like $v\_\theta $ does not guarantee the existence of a potential function. On the other hand, Theorem 3.3 proves that $v^* $ is the gradient of smoothed distance functions, which ensures $\mathcal{M}'\_t$ corresponds to the stationary points of a well-define $F\_t(x)$ with $0 = \nabla F\_t(x) \propto \nabla (f\_t^y(x) - f\_t^\emptyset(x))$. Therefore, we use $\mathcal{M}\_{t} \approx \mathcal{M}'\_{t}$ and we implement the projection onto $\mathcal{M}'\_{t}$.
>
> - While the incremental updates in (3.7) are derived using the gradients of the ideal potential $v^* $, this field is computationally intractable. We therefore use $v\_\theta \approx v^* $ to substitute $v^* $ with $v\_\theta$ to arrive at our final implementable form.
>
> **[W2] optimal settings for baseline:**
> Our experimental framework and the selected default ranges for the guidance scale and NFEs strictly follow the implementation in [1,2,3]. Even when we select the best possible metrics achieved by each method across all evaluated combinations of guidance scale and NFEs, our proposed approach remains superior. For example, for the Pick-a-Pic dataset in Table 3, we aggregate the peak performance for each method under $\omega \in \lbrace 3.0, 4.0 \rbrace$ and NFE $\in \lbrace 60, 100 \rbrace$, see Table R3:
>
> **Table R3. Best achieved metrics for SD3.5 on the Pick-a-Pic dataset (Best of $\omega \in \lbrace 3.0, 4.0 \rbrace$ and NFE $\in \lbrace 60, 100 \rbrace$).**
> | Method | Best Pick ($\uparrow$) | Best CLIP ($\uparrow$) | Best HPS ($\uparrow$) | Best IR ($\uparrow$) |
> | :--- | :---: | :---: | :---: | :---: |
> | CFG | 22.34 | 32.57 | 30.22 | 1.05 |
> | R-CFG++ | 22.37 | 32.60 | 30.45 | 1.08 |
> | CFG-0* | 22.36 | 32.61 | 30.24 | 1.07 |
> | **CFG-MP** | 22.40 | 32.64 | 30.78 | 1.11 |
> | **CFG-MP+** | **22.51** | **32.75** | **31.15** | **1.16** |
>
>
> **[Q2] the smoothing parameter in $v_\theta(t, x, y) $ and $v^\ast_{t, y}  $:**
> In our theoretical framework, we have $\sigma = 1-t$ and the model is trained to approximate the ideal velocity field ($v_\theta(t,x,y) \approx v^*_{t,y}(x)$). Therefore, both $v_\theta(t, x, y)$ and $v^\ast_{t, y}  $ capture the smooth schedule of the smoothed distance function at each timestep.
>
>
>
>
> **[Q3] approximation of $\text{dist}^2_{t\mathcal{K}_y}(x,1-t)$:**
> We clarify that the smoothing parameter $\sigma = 1-t$ is an intentional mechanism of homotopy optimization, facilitating a transition from global mean attraction to local projection (see [4] for a comprehensive analysis). 1) In the initial stage ($t \to 0$), where $\sigma \to 1$, the posterior data distribution simplifies as the exponential term becomes nearly independent of $x$:
>
> $p\_1^y(x \mid z,t) = \frac{\exp \left( -\frac{\|z-tx\|^2}{2(1-t)^2} \right) p\_1^y(x)}{\int\_{\mathbb{R}^d} \exp \left( -\frac{\|z-tx\_1\|^2}{2(1-t)^2} \right) p\_1^y(x\_1) dx\_1} \xrightarrow{t \to 0} p\_1^y(x). $
>
> Substituting this into the ideal velocity field,
>
> $v\_{t, y}^* (x) = \frac{1}{1-t} ( \mathbb{E}\_{\tilde{x}\_1 \sim p\_1^y(\cdot \mid x,t)} [\tilde{x}\_1] - x ) \xrightarrow{t \to 0} \mathbb{E}\_{x\_1 \sim p\_1^y}[x\_1] - x,$
>
> therefore the ODE $\tfrac{d}{dt}x\_t = v\_{t, y}^* (x)$ is initially attracted to the data mean. 2) In the final stage ($t \to 1$), as $\sigma \to 0$, the smoothed distance converges to the exact squared distance: $\lim\_{\sigma \to 0} \text{dist}\_{\mathcal{K}\_y}^2(x, \sigma) = \text{dist}\_{\mathcal{K}\_y}^2(x)$. In this regime, the velocity field $v^* $ converges to the exact gradient of the squared distance to the target set $\mathcal{K}\_y$, where $v\_{t, y}^* (x) \propto -\nabla\_x \text{dist}\_{\mathcal{K}\_y}^2(x)$, ensuring the trajectory performs a precise local projection.
>
> **References**
>
> [1] Esser et al. Scaling Rectified Flow Transformers for High-Resolution Image Synthesis, _ICML_, 2024.
>
>
> [2] Fan et al. CFG-Zero: Improved Classifier-Free Guidance for Flow Matching Models, _arXiv:2503.18886_, 2025.
>
> [3] Saini et al. Rectified CFG++ for Flow Based Models, _NeurIPS_, 2025
>
> [4] Wan et al. Elucidating Flow Matching ODE Dynamics via Data Geometry and Denoisers, _ICML_, 2025.

---

> > ### Author Rebuttal · Reviewer_Lj88 · 2026-04-03
> >
> > I thank the authors for the detailed rebuttal. Since my concerns, mainly regarding the approximations made in the paper, are resolved, I raise my score to 4 accordingly.

---

> > > ### Author Response · Authors · 2026-04-04
> > >
> > > Dear Reviewer Lj88,
> > >
> > > Thank you for upgrading your score and for your valuable suggestions throughout this process. We are very glad that we could address your concerns effectively.
> > >
> > > Best regards,
> > >
> > > The Authors

---

### Official Review · Reviewer_DEBQ · 2026-03-10

**Soundness:** 3
**Presentation:** 2
**Significance:** 2
**Originality:** 3
**Overall Recommendation:** 3
**Confidence:** 3

**Summary:**

This paper provides an optimization-based interpretation of Classifier-Free Guidance (CFG) in flow matching, demonstrating that the velocity field corresponds to the gradient of a smoothed distance function. The authors identify the "prediction gap" as the primary source of guidance scale sensitivity and propose CFG-MP, a training-free sampling method that incorporates iterative manifold projection to eliminate this gap.

**Compliance With Llm Reviewing Policy:**

Affirmed.

**Final Justification:**

While I acknowledge the motivation of the proposed approach, concerns remain regarding the insufficient justification of the underlying manifold assumption. In addition, the empirical evaluation relies on proxy metrics and relatively modest gains, making it unclear how strongly the improvements translate to actual generation quality. Furthermore, the introduction of additional hyperparameters raises questions about whether the method truly resolves sensitivity issues or simply redistributes them, weakening the overall claim of practical effectiveness.

**Key Questions For Authors:**

In the performance comparison, DDIM-based methods are compared against vanilla CFG (DDIM), while the proposed method is compared against CFG (D2F). This inconsistency makes it difficult to clearly distinguish whether the performance gains stem from the algorithmic contribution or simply from the framework difference. A fair comparison would require evaluating all methods under the same framework (either all DDIM or all D2F) to isolate the actual benefit of the proposed approach.

**Limitations:**

Yes

**Strengths And Weaknesses:**

## Strengths
1. The reinterpretation of CFG sampling as a homotopy optimization process that converges to the data manifold along the gradient of a normalized distance function, rather than simple numerical extrapolation, is quite interesting and provides fresh theoretical insight.
2. The interpretation of the velocity field in flow matching as a combination of attractive forces pulling toward the target image set and repulsive forces pushing away from the origin offers an intuitive understanding of the guidance mechanism.
3. The authors mathematically demonstrate that guidance sensitivity stems from the prediction gap, providing logical justification for the necessity of manifold projection. This theoretical grounding strengthens the motivation for their approach.

## Weakness
1. **Lack of mathematical guarantees for manifold existence**: The manifold $\mathcal{M}_t$ is defined as points where $v_\theta(y) = v_\theta(\emptyset)$. However, there is insufficient mathematical guarantee that such points exist near the sampling trajectory in high-dimensional spaces. The authors should provide either theoretical analysis (e.g., using implicit function theorem or topological arguments) or empirical verification that these manifold points are consistently found during sampling.

2. **Excessive hyperparameter burden**: The Anderson acceleration introduces numerous hyperparameters: window size, damping factor, number of iterations, and step size. This seems contradictory to the stated goal of alleviating the "inconvenience of CFG scale tuning"—the method essentially replaces one tunable parameter with four (or more). The practical usability is questionable without clear guidelines for setting these values across different domains.

3. **Computational overhead despite "training-free" claim**: While advertised as training-free, the method requires $K$ additional projection iterations at each sampling step, with multiple evaluations of $v_\theta$ per iteration. Although CFG-MP+ attempts to mitigate this with Anderson acceleration, the computational cost appears prohibitively high. The paper would benefit from explicit wall-clock time comparisons and NFE (number of function evaluations) analysis to assess practical feasibility, especially for large-scale generation tasks.

---

> ### Author Rebuttal · Authors · 2026-03-28
>
> Thank you for your detailed and valuable suggestions. They play a crucial role in improving our manuscript.
>
> **[W1] manifold existence:** Our objective is to reduce the prediction gap to decrease guidance scale sensitivity. The manifold $\mathcal{M}_t$ serves as a formalization for ease of presentation, but our method remains robust in its absence. As detailed in Remark 3.6, the iterative process minimizes the potential $F_t(x)$ regardless of the manifold's state, successfully narrowing the prediction gap (see Fig. 4). Consequently, a formal proof of the manifold's existence is not a prerequisite for the validity of our results and is thus omitted.
>
> **[W2] parameters in AA:**
> While our method incorporates Anderson Acceleration (AA), it does not introduce a practical tuning burden. Unlike the guidance scale $w$, which is highly sensitive and requires per-case adjustment, **the AA parameters are stable and remain fixed across all experiments.** Specifically, we employ a default setting of $m=1, \beta = 1$ throughout Sections 4.1, 4.2, 4.3 (the number of iterations $K$ and the stepsize $\Delta t$ depend totally on the practical computation budget); and this choice ($m=1, \beta = 1$ ) is formally justified by our principal analysis and empirical results.
>
> **[W3] computational cost:**
> 1) *Rigorous Comparison:* To ensure a fair comparison, all evaluated methods are constrained to the same total Number of Function Evaluations (NFEs). Our approach does not add to the computational budget; rather, it reallocates it. For example, at a fixed budget of NFE=60, the vanilla CFG baseline utilizes 30 inference steps. In contrast, CFG-MP/MP+ employ 18 inference steps with additional inner-loop iterations (the first 6 steps with $K=2$ and the last 12 with $K=0$), ensuring the total NFEs remains identical across all methods.
> 2) *Wall-clock Time:* Here, we conducted wall-clock time comparisons on Stable Diffusion 3.5 to demonstrate that the projection iterations do not impose a significant computational burden. As shown in Table R2, the generation time per image remains highly competitive across all settings, confirming the practical feasibility of CFG-MP+ for large-scale generation tasks.
>
> **Table R2. Generation time (s) per image for various sampling methods.**
> | NFE | CFG | CFG-0* | R-CFG++ | CFG-MP | CFG-MP+ |
> | :--- | :--- | :--- | :--- | :--- | :--- |
> | 60 | 15.43s | 15.67s | 15.22s | 14.98s | 15.04s |
> | 100 | 25.89s | 25.27s | 25.42s | 25.08s | 25.24s |
>
> **[Q] fair comparison:**
> 1) In Table 3, we compared all the methods in the flow matching (FM) framework across all text-to-image tasks involving FLUX.1-dev and SD3.5.
> 2) We included the comparison of some diffusion-model-based methods like Z-sampling, Re-sampling, and FSG, since they share the same objective of reducing the prediction gap (See [1]). Furthermore, we discuss the influence of the underlying framework (D2F vs. DDIM) through a relative improvement analysis in Appendix C.2.2. This analysis evaluates each method against its respective baseline, and as demonstrated in Tables 7 and 8, CFG-MP+ achieves the most significant relative improvements in both FID (robustness)  and IS (gain), confirming that the performance benefits stem from our algorithmic contribution rather than framework differences.
>
> **References**
>
> [1] Wang et al. Towards a Golden Classifier-Free Guidance Path via Foresight Fixed Point Iterations, _NeurIPS_, 2025.

---

> > ### Author Rebuttal · Reviewer_DEBQ · 2026-04-03
> >
> > Thank you for the detailed and thoughtful response. While I appreciate the clarifications, some of my concerns remain insufficiently addressed.
> >
> > [W1] The rebuttal does not address the existence or consistency of the manifold $\\mathcal{M}_t$ assumed in the formulation. Arguing that the method works without the manifold does not resolve the concern, but instead weakens the theoretical grounding of the approach.
> >
> > [W2] The claim that the choice of $m=1, \beta=1$ is “formally justified” is not sufficiently supported, as neither a clear theoretical argument nor an empirical sensitivity analysis is provided. Moreover, the method introduces additional parameters such as the number of iterations $K$ and the step size $\Delta t$, beyond $m$ and $\beta$.
> > This raises the question of whether the overall hyperparameter complexity and sensitivity is actually reduced compared to CFG. Since alleviating CFG’s sensitivity is a central motivation of the paper, a clearer and more comprehensive justification is necessary.

---

> > > ### Author Response · Authors · 2026-04-04
> > >
> > > We thank the reviewer for pushing us to clarify these critical points and realize that our previous response may not have fully addressed the core concerns.
> > >
> > > **[w1]:**
> > > The manifold $\mathcal{M}\_t$ approximates the set $\mathcal{M}' \_{t}:= \lbrace z \mid v^* \_{t,y}(z) = v^* \_{t,\emptyset}(z) \rbrace$. As established in Theorem 3.3, we can verify its existence by determining whether a point $z$ exists such that its projections onto the conditional dataset $\mathcal{K}\_y$ and the whole-image dataset $\mathcal{K}\_\emptyset$ are identical. While rigorously proving this requires specific geometric assumptions regarding the relationship between $\mathcal{K}_y$ and $\mathcal{K}\_\emptyset$, we empirically observe that this condition is met at the late stages of the trajectory (when $t$ is sufficiently large). Conversely, at early stages (when $t$ is small), this manifold serves as a conceptual target for optimization. In this regime, our method with the manifold acts as a regularizer that minimizes the prediction gap, operating effectively within an energy landscape where our algorithm takes gradient steps to pull the latent variables toward regions of lowest potential.
> > >
> > > **[w2]:**
> > > For the sensitivity analysis regarding the step size $a := \lambda \Delta t$, please refer to Tables 4 and 5 in the original appendix. These results demonstrate that performance is not sensitive to $\lambda$, justifying our choice of $\lambda = 0.5$ across these datasets. Regarding the sensitivity analysis for $(K, m, \beta)$, please refer to Tables R4 and R5 below. We observed that $(m, \beta) = (1,1)$ yields the optimal values across these datasets. Additionally, while larger values of $K$ improve performance at the expense of higher computational cost, we found that $K=2$ or $3$ allows our algorithm to outperform competing methods while maintaining the same Number of Function Evaluations (NFE).
> > >
> > >
> > > **Table R4: Sensitivity Analysis of hyperparameters from Anderson Acceleration in terms of relative change $(-r \uparrow). $**
> > > | $m$ | $\beta$ | $K=3$ | $K=4$ | $K=5$ |
> > > | :---: | :---: | :---: | :---: | :---: |
> > > | 1 |0.2 | 9.31 | 13.52 | 16.85 |
> > > | 1 | 0.4 | 9.32 | 13.54 | 16.82 |
> > > | 1 | 0.6 | 9.37 | 13.64 | 16.76 |
> > > | 1 | 0.8 | 9.36 | 13.56 | 16.74 |
> > > | 1 | 1.0 | **9.44** | **13.76** | **16.97** |
> > > | 2 |  0.2 | 9.30 | 13.49 | 16.70 |
> > > | 2 |  0.4 | 9.35 | 13.52 | 16.72 |
> > > | 2 | 0.6 | 9.33 | 13.54 | 16.75 |
> > > | 2 | 0.8 | 9.31 | 13.53 | 16.73 |
> > > | 2 | 1.0 | 9.32 | 13.54 | 16.82 |
> > > | 3|0.2 | 9.30 | 13.50 | 16.71 |
> > > | 3|0.4 | 9.29 | 13.51 | 16.73 |
> > > | 3| 0.6 | 9.31 | 13.52 | 16.74 |
> > > | 3|0.8 | 9.28 | 13.51 | 16.72 |
> > > | 3| 1.0 | 9.31 | 13.59 | 16.80 |
> > >
> > > **Table R5: Sensitivity Analysis of hyperparameters from Anderson Acceleration  in terms of image reward (IR$\uparrow$) and Human Preference Score (HPS$\uparrow$).**
> > >
> > > | $m$ | $\beta$ | $K=3$ (IR / HPS) | $K=4$ (IR / HPS) | $K=5$ (IR / HPS) |
> > > | :---: | :---: | :---: | :---: | :---: |
> > > | 1 | 0.2 | 1.08 / 30.92 | 1.11 / 31.04 | 1.13 / 31.07 |
> > > | 1 | 0.4 | 1.09 / 30.90 | 1.12 / 31.03 | 1.14 / 31.06 |
> > > | 1 | 0.6 | 1.10 / 30.96 | 1.14 / 31.08 | 1.15 / 31.09 |
> > > | 1 | 0.8 | 1.10 / 30.93 | 1.13 / 31.05 | 1.14 / 31.08 |
> > > | 1 | 1.0 | **1.12 / 31.00** | **1.15 / 31.12** | **1.18 / 31.16** |
> > > | 2 |  0.2 | 1.05 / 30.92 | 1.11 / 31.04 | 1.15 / 31.07 |
> > > | 2 | 0.4 | 1.06 / 30.90 | 1.11 / 31.02 | 1.14 / 31.05 |
> > > | 2 | 0.6 | 1.06 / 30.95 | 1.13 / 31.06 | 1.16 / 31.07 |
> > > | 2 | 0.8 | 1.07 / 30.94 | 1.12 / 31.07 | 1.15 / 31.11 |
> > > | 2 | 1.0 | 1.09 / 30.96 | 1.13 / 31.10 | 1.16 / 31.13 |
> > > | 3 | 0.2 | 1.07 / 30.92 | 1.09 / 31.05 | 1.14 / 31.08 |
> > > | 3 | 0.4 | 1.07 / 30.91 | 1.10 / 31.07 | 1.13 / 31.12 |
> > > | 3 |  0.6 | 1.09 / 30.95 | 1.12 / 31.06 | 1.15 / 31.11 |
> > > | 3 |  0.8 | 1.08 / 30.94 | 1.11 / 31.05 | 1.14 / 31.09 |
> > > | 3 | 1.0 | 1.09 / 30.95 | 1.12 / 31.09 | 1.15 / 31.14 |

---

### Official Review · Reviewer_hPjZ · 2026-03-13

**Soundness:** 4
**Presentation:** 3
**Significance:** 3
**Originality:** 3
**Overall Recommendation:** 5
**Confidence:** 2

**Summary:**

The paper studies classifier-free guidance (CFG) in flow matching from an optimization perspective, arguing that sampling follows a homotopy path over smoothed distance objectives. It identifies prediction gap between conditional and unconditional velocity fields as a key sensitivity amplifier for guidance scale, then proposes CFG-MP (manifold projection) and CFG-MP+ (with Anderson acceleration) as training-free corrections. The empirical section reports improved robustness and stronger semantic alignment/compositionality across representative flow-matching backbones. Overall, this is a technically meaningful and practically motivated contribution that advances understanding of inference-time control in flow matching.

**Compliance With Llm Reviewing Policy:**

Affirmed.

**Final Justification:**

Thank the authors for their response for addressing my concerns. I would like to maintain my positive score.

**Key Questions For Authors:**

1. How sensitive is performance to the smoothing schedule choice beyond sigma=1-t, and can adaptive schedules improve stability?
2. Can you provide diagnostics/predictors for projection failure or oscillation (especially in high-curvature regions)?
3. How does CFG-MP+ perform under conflicting conditioning signals (e.g., positive+negative prompts or compositional attribute conflict)?
4. Can the method preserve diversity while reducing prediction gap? Please report diversity-sensitive metrics or analyses.

**Limitations:**

Yes. The paper acknowledges stability/efficiency trade-offs of iterative projection and partially addresses them with Anderson acceleration. A more explicit failure-case section would further improve transparency.

**Strengths And Weaknesses:**

**Strengths**:

* Clear and valuable reframing of CFG as path-tracking in a time-varying optimization landscape, rather than a purely heuristic extrapolation.

* The error decomposition connecting guidance mismatch to the prediction gap is insightful and helps explain scale sensitivity in a principled way.

* CFG-MP/CFG-MP+ are training-free and therefore useful for large pre-trained models where retraining is costly.

* Experiments span multiple modern backbones and include both distributional and preference/compositional indicators, which is appropriate for the claims.

**Weaknesses**:

* Theoretical assumptions around smoothing schedule and manifold geometry may not fully match real high-dimensional latent landscapes; failure conditions are not yet characterized sharply.

* Some gains appear metric-dependent and there remains potential tension between stronger conditioning and global distribution coverage/diversity.

* The method is mostly evaluated under single-condition settings; behavior under conflicting multi-condition prompts (including negative prompts) needs deeper analysis.

---

> ### Author Rebuttal · Authors · 2026-03-28
>
> **[W1] assumptions on the smoothing schedule and the manifold, as well as the failure characterization**:
>
> 1) Regarding the smoothing schedule, **we do not impose any new assumption.**  The smoothing parameter $\sigma = 1-t$ is directly from the flow matching (FM) loss functional (See Theorem 3.1 and 3.3). As $t \to 1$ and $\sigma \to 0$, our smoothed distance function provably converges to the true squared distance to the target image set. This schedule is therefore an intrinsic property of the FM transition from noise to data, rather than a theoretical assumption.
>
> 2) Regarding the manifold $\mathcal{M}_t$, we assume $\mathcal{M}_t$ is not empty. We clarify that this assumption is not our end goal. Rather, our objective is to reduce the prediction gap to decrease guidance scale sensitivity. The manifold $\mathcal{M}_t$ serves as a formalization for ease of presentation, but our method remains robust in its absence. As detailed in Remark 3.6, the iterative process minimizes the potential $F_t(x)$ regardless of the manifold's state, successfully narrowing the prediction gap (see Fig. 4).
>
>
> 3) For the failure condition characterization, see **[Q2]** below for a detailed discussion.
>
>
>
> **[W2 & Q4] potential tension between stronger conditioning and global diversity**:
> As demonstrated in Table 6, this tension does not appear to be a limiting factor for our approach. While the CFG-D2F baseline achieves the lowest FID (a diversity-sensitive metric) in two specific high-NFE scenarios ($\omega=1.5$ and $\omega=2.0$ at NFE=120), our methods CFG-MP and CFG-MP+ achieve the best FID in the other 4 evaluated groups. Furthermore, CFG-MP+ consistently yields the highest IS (a conditioning-sensitive metric) across all evaluated guidance scales and NFE budgets. Empirical results (Figure 8) further demonstrate that CFG-MP/MP+ do not suffer from common issues such as loss of diversity.
>
> **[W3 & Q3] conflicting conditioning signals:**
> Our framework is natively compatible with standard CFG techniques, including the use of negative prompts $y^{-}$. Mathematically, even when conditioning signals $y^{+}$ and $y^{-}$ are conflicting, the operator
> $G(x,t) \coloneqq x - a\cdot v_{\theta}(t,x,y^{-}) + a\cdot v_{\theta}\left(t,x - a\cdot v_{\theta}(t,x,y^{-}),y^{+} \right)$ continues to function as an optimization step.
> By minimizing the prediction gap $||v_{\theta}^{+} - v_{\theta}^{-}||_2$ between these conflicting signals, the operator effectively mitigates the sensitivity to the guidance scale $w$, ensuring stable generation regardless of the divergence between prompts.（The designs of conflicting $y^{+}$ and $y^{-}$ comes from [1]）
>
> **Table R1. Performance Comparison of CFG Methods under Null and Negative Prompts, $\omega = 3$ and NFE $=60$.**
> | Method | Null Prompt (HPS) | Null Prompt (IR) | Negative Prompt (HPS) | Negative Prompt (IR) |
> | :--- | :---: | :---: | :---: | :---: |
> | CFG | 29.76 | 1.00 | 29.87 | 1.02 |
> | CFG-MP | 29.77 | 1.03 | 30.11 | 1.06 |
> | **CFG-MP+** | **30.78** | **1.11** | **30.94** | **1.17** |
>
> **[Q1] Sensitivity of smoothing schedule $\sigma = 1-t$**:
> We clarify that the choice of $\sigma = 1-t$ in Theorem 3.3 is not heuristic; it is derived directly from the minimizer of the training objective of CFG in flow matching (Theorem 3.1). This unique, intrinsic smoothing schedule ensures that the learned velocity field $v_{\theta}$ corresponds precisely to the gradient of the smoothed distance function in homotopy optimization. Altering this schedule would create a mathematical mismatch between the model's learned velocity field and the inference-time optimization. Therefore, any deviation from the $\sigma = 1 - t$ schedule leads to **the requirement of model retraining**, which falls outside the scope of our training-free framework.
>
> **[Q2] diagnostics for projection failure:**
> In Section 4.3, we utilize the relative change $r$ as a principled diagnostic tool to monitor the convergence of the projection process in real time. Our analysis reveals that vanilla fixed-point iterations of $G(x,t)$ (CFG-MP) are prone to instability and divergence during the transition between the early and middle sampling stages. This observation was the key motivation for developing CFG-MP+. By incorporating Anderson Acceleration (AA), we effectively stabilize these non-contractive operators, ensuring the prediction gap decays reliably even in high-curvature regions where standard manifold projection might fail.
>
> **References**
>
> [1] Ban et al. Understanding the Impact of Negative Prompts: When and How Do They Take Effect?, _ECCV_, 2024.

---

> > ### Author Rebuttal · Reviewer_hPjZ · 2026-04-04
> >
> > I thank the authors for their thorough response. I would like to maintain my positive score.

---

> > > ### Author Response · Authors · 2026-04-04
> > >
> > > Dear Reviewer hPjZ,
> > >
> > > We thank the reviewer for their careful review and constructive feedback. We also appreciate the acknowledgment that our rebuttal has fully addressed the concerns raised in the initial review.
> > >
> > > Best regards,
> > >
> > > The Authors

---

### Official Review · Reviewer_LYNS · 2026-04-07

**Soundness:** 3
**Presentation:** 3
**Significance:** 3
**Originality:** 3
**Overall Recommendation:** 4
**Confidence:** 4

**Summary:**

The paper aims to improve classifier-free guidance of flow matching. The approach of the paper is to view the flow matching with CFG as an optimization problem where the objective is to minimize the distance between generated output and target. Given the error caused by CFG, the use of fix $w$ resulted in amplifying those errors. In order to solve this problem, the author proposed to project the intermediate point onto a new manifold so that the error between conditional velocity and unconditional velocity is minimized. By doing so, the authors argue that it helped to correct the error caused by cfg, hence it will work with different values of w.

**Compliance With Llm Reviewing Policy:**

Affirmed.

**Final Justification:**

Will adjust the score if the rebuttal already addressed the concerns.

**Key Questions For Authors:**

Please see the weaknesses

**Strengths And Weaknesses:**

Strength:
1. The idea of projection on manifold is novel
2. The theoretical parts of the paper are well written
3. The structure of the paper is clear

Weaknesses:
1. Computational cost is surely very high for work. CFG is well-known for its expensiveness where it have to double the cost of sampling. This work proposes to use K times CFG which K is a hyperparameter. This challenges the practical of this work.
2. There is no guaranteed bound for $K$ to do projection on the defined manifold. Choosing $K$ should be a trade-off between quality and sampling time.
3. Given the sampling time is very high, the performance is slightly improved. This would discourage people to adapt this method into the work.

---

### Decision · Program_Chairs · 2026-04-30

**Decision:**

Accept (regular)

**Comment:**

The paper studies classifier free guidance. Leveraging an optimization standpoint, CFG is presented as an homotopy path over a smoothed distance. Following [1], a "prediction gap" is identified, namely the difference between conditional and unconditional velocities; iterated projections onto the manifolds of points where the unconditional and conditional velocities are used to reduce it.

Reviewer `hPjZ` highlights the interesting optimization perspective on CFG. So does reviewer `Lj88`, who was satisfied with authors rebuttal and also appreciated the proposed way to reduce the prediction gap.
Reviewer `DEBQ` was critical of the additional hyperparameters induced by Anderson Acceleration, but the latter is a fairly frequent tool that often performs well across a large range of default values. As pointed out by `DEBQ` and `LYNS` the method requires $K$ additional projection iterations at each sampling step, but experiments provided by the authors with a fixed NFE budget across competitors still demonstrated the interest of the method.

[1] Towards a golden classifier-free guidance path via foresight fixed point iterations, Wang et al, Neurips 2025.